# Genetic correlations and genome-wide associations of cortical structure in general population samples of 22,824 adults

Edith Hofer et al.[#]

Cortical thickness, surface area and volumes vary with age and cognitive function, and in neurological and psychiatric diseases. Here we report heritability, genetic correlations and genome-wide associations of these cortical measures across the whole cortex, and in 34 anatomically predefined regions. Our discovery sample comprises 22,824 individuals from 20 cohorts within the Cohorts for Heart and Aging Research in Genomic Epidemiology (CHARGE) consortium and the UK Biobank. We identify genetic heterogeneity between cortical measures and brain regions, and 160 genome-wide significant associations pointing to wnt/β-catenin, TGF-β and sonic hedgehog pathways. There is enrichment for genes involved in anthropometric traits, hindbrain development, vascular and neurodegenerative disease and psychiatric conditions. These data are a rich resource for studies of the biological mechanisms behind cortical development and aging.

[#]A list of authors and their affiliations appears at the end of the paper.

The cortex is the largest part of the human brain, associated with higher brain functions, such as perception, thought, and action. Brain cortical thickness (CTh), cortical surface area (CSA), and cortical volume (CV) are morphological markers of cortical structure obtained from magnetic resonance imaging (MRI). These measures change with age[1-3] and are linked to cognitive functioning[4,5]. The human cortex is also vulnerable to a wide range of disease or pathologies, ranging from developmental disorders and early onset psychiatric and neurological diseases to neurodegenerative conditions manifesting late in life. Abnormalities in global or regional CTh, CSA, and CV have been observed in neurological and psychiatric disorders, such as Alzheimer's disease[6], Parkinson's disease[7], multiple sclerosis[8], schizophrenia[9], bipolar disorder[9], depression[10], and autism[11]. The best method to study human cortical structure during life is using brain MRI. Hence, understanding the genetic determinants of the most robust MRI cortical markers in apparently normal adults could identify biological pathways relevant to brain development, aging, and various diseases. Neurons in the neocortex are organized in columns which run perpendicular to the surface of the cerebral cortex[12]; and, according to the radial unit hypothesis, CTh is determined by the number of cells within the columns and CSA is determined by the number of columns[13].

Thus, CTh and CSA reflect different mechanisms in cortical development[13,14] and are likely influenced by different genetic factors[15-18]. CV, which is the product of CTh and CSA, is determined by a combination of these two measures, but the relative contribution of CTh and CSA to CV may vary across brain regions. CTh, CSA, and CV are all strongly heritable traits[15-21] with estimated heritability of 0.69–0.81 for global CTh, and from 0.42 to 0.90 for global CSA[15,16,18]. Across different cortical regions, however, there is substantial regional variation in heritability of CTh, CSA, and CV[15-21].

Since CTh, CSA, and CV are differentially heritable and genetically heterogeneous, we explore the genetics of each of these imaging markers using genome-wide association analyses (GWAS) in large population-based samples. We study CTh, CSA, and CV in the whole cortex and in 34 cortical regions in 22,824 individuals from 21 discovery cohorts and replicate the strongest associations in 22,363 persons from the Enhancing Neuroimaging Genetics through Meta-analysis (ENIGMA) consortium. Our analyses reveal 160 genome-wide significant associations pointing to wnt/β-catenin, TGF-β, and sonic hedgehog pathways. We observe genetic heterogeneity between cortical measures and brain regions and find enrichment for genes involved in anthropometric traits, hindbrain development, vascular and neurodegenerative disease, and psychiatric conditions.

## Results

**Genome-wide association analysis**. The analyses of global CTh, CSA, and CV included 22,163, 18,617, and 22,824 individuals, respectively. After correction for multiple testing ($p_{Discovery} < 1.09 \times 10^{-9}$), we identified no significant associations with global CTh. However, we identified 12 independent loci associated with global CSA ($n = 6$) and CV ($n = 6$). These are displayed in Supplementary Data 1 and Supplementary Figs. 1 and 2. Five of the 6 CSA loci were replicated in an external (ENIGMA consortium) sample[22]. The ENIGMA consortium only analyzed CSA and CTh.

GWAS of CTh, CSA, and CV in 34 cortical regions of interest (ROIs) identified 148 significant associations. There were 16 independent loci across 8 chromosomes determining CTh of 9 regions (Supplementary Data 2), 54 loci across 16 chromosomes associated with CSA of 21 regions (Supplementary Data 3), and 78 loci across 17 chromosomes determining CV of 23 cortical

regions (Supplementary Data 4). We replicated 57 out of 64 regional CTh and CSA loci that were available in the ENIGMA consortium sample[22] using a conservative replication threshold of $p_{Replication} = 3.1 \times 10^{-4}$, 0.05/160. Region-specific variants with the strongest association at each genomic locus are shown in Tables 1–3. Chromosomal ideograms showing genome-wide significant associations with global and regional cortical measures in the discovery stage are presented in Fig. 1.

If we had used a more stringent threshold of $p_{Discovery} < 4.76 \times 10^{-10} = 5 \times 10^{-8}/105$, correcting for all the 105 GWAS analyses performed, we would have identified 142 significant associations (Supplementary Data 1–4).

The strongest associations with CTh and CV were observed for rs2033939 at 15q14 ($p_{Discovery, CTh} = 1.17 \times 10^{-73}$ and $p_{Discovery, CV} = 4.34 \times 10^{-133}$) in the postcentral (primary somatosensory) cortex, and for CSA with rs1080066 at 15q14 ($p_{Discovery, CSA} = 8.45 \times 10^{-109}$) in the precentral (primary motor) cortex. Figure 2 shows the lowest p-value of each cortical region. The postcentral cortex was also the region with the largest number of independent associations, mainly at a locus on 15q14. The corresponding regional association plots are presented in Supplementary Fig. 3. Quantile-quantile plots of all meta-analyses are presented in Supplementary Figs. 4–7 and the corresponding genomic inflation factors ($\lambda_{GC}$), LD score regression (LDSR) intercepts, and ratios are shown in Supplementary Data 5. Although we observe inflated test statistics for some traits with $\lambda_{GC}$ between 1.02 and 1.11, LDSR intercepts between 0.98 and 1.02 indicate that the inflation is mainly due to polygenicity. For traits with $\lambda_{GC} > 1.05$, the LDSR ratios range between 0.00 and 0.15 which means that a maximum of 15% of the inflation is due to other causes.

**Associations across cortical measures and with other traits**. Supplementary Data 6 presents variants that are associated with the CSA or the CV across multiple regions. We observed 25 single nucleotide polymorphisms (SNPs) that determined both the CSA and CV of a given region, 4 SNPs that determined CTh and CV of the same region, but no SNPs that determined both the CTh and CSA of any given region (Supplementary Data 7). We also checked the overlap between our findings and two previous GWAS studies, including 8428[23] and 19,621[24] individuals from the UK Biobank, which among other phenotypes, investigate CTh, CSA, and CV (Supplementary Data 8). Regarding CTh, one variant, rs2033939 at 15q14, was associated with CTh of the postcentral gyrus in both studies. For CSA and CV, we found 11 associations at 15q14, 14q23.1 and 3q24, and 14 associations at 15q14, 14q23.1, 3q24, 8q24.1, 12q14.3, and 20q13.2, respectively, with the same cortical region as in our study. Out-of-sample polygenic risk score (PRS) analyses showed associations ($p_{PRS} < 4.76 \times 10^{-3}$) with all investigated cortical measures in all cortical regions in 7800 UK Biobank individuals (Supplementary Data 9). For CTh, we observed the maximum phenotypic variance explained by the PRS ($R_{PRS}^2$) in the global cortex ($R_{PRS}^2 = 0.015$, $p_{PRS} = 1.05 \times 10^{-26}$), and for CSA and CV in the pericalcarine cortex ($R_{PRS}^2$,CSA = 0.029, $p_{PRS,CSA} = 1.29 \times 10^{-50}$; $R_{PRS}^2$,CV = 0.032, $p_{PRS,CV} = 5.30 \times 10^{-56}$). When assessing genetic overlap with other traits, we observed that SNPs determining these cortical measures have been previously associated with anthropometric (height), neurologic (Parkinson's disease, corticobasal degeneration, and Alzheimer's disease), psychiatric (neuroticism and schizophrenia) and cognitive performance traits as well as with total intracranial volume (TIV) on brain MRI (Supplementary Data 10–12).

**Gene identification**. Positional mapping based on ANNOVAR showed that most of the lead SNPs were intergenic and intronic

**Table 1 Genome-wide significant associations ($p_{Discovery}$ < 1.09 × 10$^{-9}$) of regional CTh.**

| Lobe | Region | Locus | Position | Lead SNP | Nearest gene | Annotation | N | $p_{Discovery}$ | $p_{Replication}$ | $p_{pooled}$ |
|---|---|---|---|---|---|---|---|---|---|---|
| Temporal | Superior temporal | 16q24.2 | 87225139 | rs4843227 | LOC101928708 | Intergenic | 21,887 | 2.79E−12 | 2.45E−05 | 2.31E−15 |
| | Middle temporal | 17q21.31 | 44861003 | rs199504 | WNT3 | Intronic | 21,887 | 1.30E−10 | 1.17E−04 | 5.85E−13 |
| | Inferior temporal | 14q23.1 | 59072144 | rs10782438 | KIAA0586 | Intergenic | 21,559 | 2.17E−13 | 2.76E−08 | 8.99E−21 |
| | Banksts | 2q35 | 217332057 | rs284532 | SMARCAL1 | Intronic | 21,885 | 1.03E−09 | 2.64E−01 | 3.04E−07 |
| Parietal | Superior parietal | 14q23.1 | 59074878 | rs160458 | KIAA0586( | Intergenic | 18,342 | 9.39E−10 | 2.42E−09 | 6.45E−18 |
| | | 16q24.2 | 87225101 | rs9937293 | LOC101928708 | Intergenic | 21,886 | 2.68E−14 | 1.64E−13 | 2.27E−27 |
| | Postcentral | 1q41 | 215141570 | rs10494988 | KCNK2 | Intergenic | 21,885 | 2.60E−12 | 3.66E−08 | 2.63E−19 |
| Occipital | Lateral occipital | 15q14 | 39633904 | rs2033939 | C15orf54 | Intronic | 21,885 | 1.17E−73 | 5.18E−68 | 7.73E−136 |
| | Cuneus | 5q14.1 | 79933093 | rs245100 | DHFR | Intergenic | 21,886 | 2.68E−11 | 3.77E−06 | 1.16E−15 |
| | | 14q23.1 | 59624317 | rs4901904 | DAAM1 | Intergenic | 21,885 | 4.02E−14 | 3.17E−10 | 2.88E−23 |
| | Insula | 16q12.1 | 51449978 | rs7197215 | SALL1 | Intergenic | 21,560 | 1.45E−13 | 2.00E−02 | 6.42E−12 |
| | | 9q31.3 | 113679617 | rs72748157 | LPAR1 | Intronic | 21,560 | 1.46E−10 | 1.38E−04 | 5.16E−13 |

N number of individuals in meta-analysis, $p_{Discovery}$ two-sided p-value of discovery GWAS meta-analysis in CHARGE, $p_{Replication}$ two-sided p-value of replication meta-analysis in ENIGMA, $p_{pooled}$ two-sided p-value of pooled discovery and replication meta-analysis, p-values are not adjusted for multiple comparisons, *bankssts* banks of the superior temporal sulcus.
in bold: significant replication—$p_{Replication}$ < 3.1 × 10$^{-4}$ (= 0.05/NI, NI=160, total number of lead SNPs).

(Fig. 3). One variant, rs2279829, which was associated with both CSA and CV of the pars triangularis, postcentral and supra-marginal cortices, is located in the 3′UTR of *ZIC4* at 3q24. We also found an exonic variant, rs10283100, in gene *ENPP2* at 8q24.12 associated with CV of the insula.

We used multiple strategies beyond positional annotation to identify specific genes implicated by the various GWAS associated SNPs. FUMA identified 232 genes whose expression was determined by these variants (eQTL) and these and other genes implicated by chromatin interaction mapping are shown in Supplementary Data 13–15. MAGMA gene-based association analyses revealed 70 significantly associated ($p < 5.87 \times 10^{-8}$) genes (Supplementary Data 16–18). For global CSA and CV, 7 of 9 genes associated with each measure overlapped, but there was no overlap with global CTh. For regional CSA and CV, we found 28 genes across 13 cortical regions that determined both measures in the same region. Figure 4 summarizes the results of GTEx eQTL, chromatin interaction, positional annotation, and gene-based mapping strategies for all regions. While there are overlapping genes identified using different approaches, only *DAAM1* (Chr14q23.1) is identified by all types of gene mapping for CV of insula. eQTL associations of our independent lead SNPs in the Religious Orders Study Memory and Aging Project (ROSMAP) dorsolateral frontal cortex gene expression dataset are presented in Supplementary Data 19.

**Pathway analysis**. MAGMA gene set analyses identified 7 pathways for CTh, 3 pathways for CSA and 9 pathways for CV (Supplementary Data 20). Among them are the gene ontology (GO) gene sets hindbrain morphogenesis (strongest association with thickness of middle temporal cortex), forebrain generation of neurons (with surface area of precentral cortex), and central nervous system neuron development (with volume of transverse temporal cortex). However, after Bonferroni correction only one significant pathway ($p < 1.02 \times 10^{-7}$) remained: regulation of catabolic process for CTh of the inferior temporal cortex. Inna-teDB pathway analyses of genes mapped to independent lead SNPs by FUMA showed a significant overlap between CTh and CSA genes and the Wnt signaling pathway (Supplementary Figs. 8 and 9) as well as a significant overlap between CV genes and the basal cell carcinoma pathway (Supplementary Fig. 10).

**Heritability**. Heritability estimates ($h^2$) of global CTh were 0.64 (standard error (se) = 0.12; $p_{SOLAR} = 3 \times 10^{-7}$) in the ASPS-Fam study and 0.45 (se = 0.08; $p_{GCTA} = 2.5 \times 10^{-7}$) in the Rotterdam study (RS). For CSA, $h^2$ was 0.84 (se = 0.12; $p_{SOLAR} = 2.63 \times 10^{-11}$) in ASPS-Fam and 0.33 (se = 0.08, $p_{GCTA} = 1 \times 10^{-4}$) in RS, and for CV, $h^2$ was 0.80 (se = 0.11; $p_{SOLAR} = 1.10 \times 10^{-9}$) in ASPS-Fam and 0.32 (se = 0.08; $p_{GCTA} = 1 \times 10^{-4}$) in RS. There was a large range in heritability estimates of regional CTh, CSA, and CV (Supplementary Data 21).

Heritability based on common SNPs as estimated with LDSR was 0.25 (se = 0.03) for global CTh, 0.29 (se = 0.04) for global CSA and 0.30 (se = 0.03) for global CV. LDSR heritability estimates of regional CTh, CSA, and CV are presented in Supplementary Data 21 and Supplementary Fig. 11. For the regional analyses, the estimated heritability ranged from 0.05 to 0.18 for CTh, from 0.07 to 0.36 for CSA and from 0.06 to 0.32 for CV. Superior temporal cortex ($h^2_{CTh} = 0.18$, $h^2_{CSA} = 0.30$, $h^2_{CV} = 0.26$), precuneus ($h^2_{CTh} = 0.16$, $h^2_{CSA} = 0.29$, $h^2_{CV} = 0.28$) and pericalcarine ($h^2_{CTh} = 0.15$, $h^2_{CSA} = 0.36$, $h^2_{CV} = 0.32$) are among the most genetically determined regions.

The results of partitioned heritability analyses for global and regional CTh, CSA, and CV with functional annotation and additionally with cell-type-specific annotation are presented in

**Table 2 Genome-wide significant associations ($p_{Discovery}$ < 1.09 × 10$^{-9}$) of global and regional CSA.**

| Lobe | Region | Locus | Position | Lead SNP | Nearest gene | Annotation | N | $p_{Discovery}$ | $p_{Replication}$ | $p_{pooled}$ |
|---|---|---|---|---|---|---|---|---|---|---|
| | Global | 17q21.31 | 44787313 | rs538628 | NSF | Intronic | 18,617 | 1.78E−23 | **4.45E−22** | 1.00E−43 |
| | | 6q22.32 | 126792095 | rs11759026 | MIR588 | Intergenic | 18,617 | 5.21E−22 | **1.45E−14** | 3.50E−34 |
| | | 6q22.33 | 127204623 | rs9375477 | RSPO3 | Intergenic | 18,617 | 4.86E−13 | **1.60E−08** | 1.23E−19 |
| | | 6q21 | 109000316 | rs9398173 | FOXO3 | Intronic | 18,617 | 6.84E−10 | 2.96E−03 | 2.05E−10 |
| | | 5q14.3 | 92187932 | rs17669337 | NR2F1-AS1 | Intergenic | 18,272 | 1.40E−11 | **2.05E−06** | 8.07E−16 |
| | | 6q22.32 | 126876580 | rs9388500 | RSPO3 | Intergenic | 17,891 | 2.35E−11 | NA | NA |
| | | 5q23.3 | 128734008 | rs12187568 | ADAMTS19 | Intergenic | 16,632 | 1.19E−16 | NA | NA |
| | | 3q24 | 147106319 | rs2279829 | ZIC4 | UTR3 | 18,265 | 6.32E−20 | **1.94E−27** | 1.20E−45 |
| | | 7q21.3 | 96175094 | rs10458281 | LOC100506136 | Intergenic | 18,265 | 1.15E−17 | **2.42E−11** | 1.20E−26 |
| | | 15q14 | 39634222 | rs1080066 | C15orf54 | Intergenic | 18,267 | 8.45E−109 | **2.53E−95** | 1.00E−200 |
| | | 6q15 | 92002569 | rs9345124 | MAP3K7 | Intergenic | 18,267 | 5.50E−11 | **2.73E−14** | 9.91E−24 |
| Frontal | Superior frontal | 2p16.3 | 48274592 | rs386645843 | FBXO11 | Intergenic | 18,269 | 9.51E−12 | **8.42E−07** | 1.71E−16 |
| | Caudal middle frontal | 4q26 | 119249835 | rs55699931 | PRSS12 | Intronic | 18,269 | 2.08E−11 | 2.72E−02 | 6.96E−10 |
| | Pars opercularis | 2q23.2 | 150022681 | rs13008194 | LYPD6B | Intronic | 18,269 | 5.94E−11 | **2.54E−07** | 1.92E−16 |
| | Pars triangularis | 6q22.32 | 126964510 | rs4273712 | RSPO3 | Intergenic | 18,269 | 6.93E−10 | **1.07E−04** | 1.99E−12 |
| | | 14q23.1 | 59072226 | rs186347 | KIAA0586 | Intergenic | 18,265 | 4.11E−10 | **1.83E−09** | 4.93E−18 |
| | Precentral | 17q21.31 | 44822662 | rs199535 | NSF | Intronic | 18,269 | 1.01E−13 | **1.14E−06** | 8.13E−18 |
| Temporal | Superior temporal | 2q23.2 | 150012936 | rs2046268 | LYPD6B | Intronic | 18,264 | 9.09E−10 | **3.21E−10** | 1.78E−18 |
| | | 15q14 | 39632013 | rs71471500 | C15orf54 | Intergenic | 18,270 | 3.85E−24 | **5.55E−19** | 5.88E−41 |
| | Middle temporal | 19p13.2 | 13109763 | rs68175985 | NFIX | Intronic | 17,324 | 8.84E−11 | **2.68E−17** | 2.90E−26 |
| | Banksts | 20q13.2 | 52448936 | rs6097618 | SUMO1P1 | Intergenic | 18,267 | 1.78E−16 | NA | NA |
| | Fusiform | 12q14.3 | 65797096 | rs2336713 | MSRB3 | Intronic | 18,267 | 1.24E−12 | **2.99E−12** | 2.85E−23 |
| | Transverse temporal | 2p25.2 | 4563477 | rs669952 | LINC01249 | Intergenic | 18,267 | 4.47E−10 | **1.37E−08** | 4.73E−17 |
| Parietal | Superior parietal | 15q14 | 39633904 | rs2033939 | C15orf54 | Intergenic | 18,272 | 9.07E−27 | **1.61E−28** | 1.59E−53 |
| | Inferior parietal | 14q23.1 | 59627631 | rs2164950 | DAAM1 | Intergenic | 18,272 | 1.25E−13 | **3.79E−14** | 3.46E−26 |
| | | 3q24 | 147106319 | rs2279829 | ZIC4 | UTR3 | 18,272 | 7.38E−12 | **4.24E−16** | 2.29E−26 |
| | Supramarginal | 15q14 | 39634222 | rs1080066 | C15orf54 | Intergenic | 18,265 | 5.65E−47 | **2.44E−36** | 1.87E−80 |
| | | 3q24 | 147106319 | rs2279829 | ZIC4 | UTR3 | 18,265 | 1.90E−21 | **1.69E−26** | 2.92E−46 |
| | Postcentral | 9q21.13 | 76144318 | rs67286026 | ANXA1 | Intergenic | 18,265 | 3.58E−12 | **8.04E−06** | 7.82E−16 |
| | | 14q23.1 | 59628609 | rs74826997 | DAAM1 | Intergenic | 18,270 | 2.40E−24 | **4.41E−18** | 4.59E−40 |
| | Precuneus | 6q23.3 | 138866268 | rs9376354 | NHSL1 | Intronic | 18,270 | 7.80E−13 | **4.12E−08** | 7.28E−19 |
| | | 3q26 | 190666643 | rs1159211 | SNAR-I | Intergenic | 18,270 | 4.49E−10 | **2.04E−05** | 1.59E−13 |
| Occipital | Lateral occipital | 14q23.1 | 59627631 | rs2164950 | DAAM1 | Intergenic | 18,269 | 3.04E−26 | **2.92E−15** | 2.25E−38 |
| | Lingual | 14q23.1 | 59628679 | rs76341705 | DAAM1 | Intergenic | 18,270 | 1.57E−20 | **8.67E−13** | 9.96E−31 |
| | Cuneus | 14q23.1 | 59625997 | rs73313052 | DAAM1 | Intergenic | 18,267 | 1.90E−32 | **3.19E−15** | 2.96E−43 |
| | | 13q31.1 | 80191873 | rs9545155 | LINC01038 | Intergenic | 18,267 | 5.15E−10 | **2.98E−05** | 3.91E−13 |
| | Pericalcarine | 14q23.1 | 59628679 | rs76341705 | DAAM1 | Intergenic | 18,267 | 4.67E−24 | **2.56E−19** | 3.35E−41 |
| | | 5q12.1 | 60117723 | rs6893642 | ELOVL7 | Intronic | 18,267 | 1.40E−13 | **1.68E−08** | 6.29E−20 |
| | | 3q13.11 | 104724787 | rs971550 | ALCAM | Intergenic | 18,267 | 2.18E−10 | **1.31E−06** | 4.49E−15 |
| | | 6q22.33 | 127185801 | rs9375476 | RSPO3 | Intergenic | 18,267 | 2.20E−10 | **2.24E−08** | 4.32E−17 |
| | | 1p13.2 | 113239478 | rs2999158 | MOV10 | Intronic | 18,267 | 6.46E−10 | **8.39E−10** | 3.49E−18 |
| | | 13q31.1 | 80191873 | rs9545155 | LINC01068 | Intergenic | 18,267 | 7.51E−10 | **7.53E−09** | 4.05E−17 |
| | Posterior cingulate | 5q12.3 | 66104105 | rs17214309 | MAST4 | Intronic | 18,268 | 7.84E−11 | **1.52E−05** | 4.04E−14 |
| | Insula | 10q25.3 | 118704077 | rs1905544 | SHTN1 | Intronic | 17,599 | 4.06E−12 | 3.65E−03 | 1.28E−11 |

N number of individuals in meta-analysis, $p_{Discovery}$ two-sided p-value of discovery GWAS meta-analysis in CHARGE, $p_{Replication}$ two-sided p-value of replication meta-analysis in ENIGMA, $p_{pooled}$ two-sided p-value of pooled discovery and replication meta-analysis, p-values are not adjusted for multiple comparisons, bankts banks of the superior temporal sulcus.
NA, SNP or region not available in the ENIGMA sample.
In bold: significant replication—$p_{Replication}$ < 3.1 × 10$^{-4}$ (= 0.05/NI, NI = 160, total number of lead SNPs).

**Table 3 Genome-wide significant associations ($p_{Discovery} < 1.09 \times 10^{-9}$) of global and regional CV.**

| Lobe | Region | Locus | Position | Lead SNP | Nearest gene | Annotation | N | $p_{Discovery}$ |
|---|---|---|---|---|---|---|---|---|
| | Global | 6q22.32 | 126792095 | rs11759026 | MIR588 | Intergenic | 22,410 | 6.31E−19 |
| | | 17q21.31 | 44790203 | rs169201 | NSF | Intronic | 22,784 | 2.11E−13 |
| | | 17q21.32 | 43549608 | rs149366495 | PLEKHM1 | Intronic | 22,099 | 8.18E−13 |
| | | 12q14.3 | 66358347 | rs1042725 | HMGA2 | 3'UTR | 22,784 | 7.04E−11 |
| | | 12q23.2 | 102921296 | rs11111293 | IGF1 | Intergenic | 22,784 | 5.45E−10 |
| | | 6q22 | 109002042 | rs4945816 | FOXO3 | 3'UTR | 22,784 | 8.93E−10 |
| Frontal | Superior frontal | 5q14.3 | 92186429 | rs888814 | NR2F1-AS1 | Intergenic | 22,692 | 3.29E−13 |
| | Rostral middle frontal | 15q14 | 39636227 | rs17694988 | C15orf54 | Intergenic | 22,793 | 3.15E−11 |
| | Caudal middle frontal | 2q12.1 | 105460333 | rs745249 | LINC01158 | ncRNA_intronic | 22,726 | 2.35E−11 |
| | | 6q22.32 | 127068983 | rs853974 | RSPO3 | Intergenic | 22,351 | 4.82E−11 |
| | Pars opercularis | 5q23.3 | 128734008 | rs12187568 | ADAMTS19 | Intergenic | 20,753 | 4.27E−18 |
| | | 15q14 | 39639898 | rs4924345 | C15orf54 | Intergenic | 22,758 | 1.97E−14 |
| | Pars triangularis | 3q24 | 147106319 | rs2279829 | ZIC4 | UTR3 | 22,759 | 3.16E−23 |
| | | 7q21.3 | 96196906 | rs67055449 | LOC100506136 | Intergenic | 22,759 | 4.03E−19 |
| | | 15q14 | 39633904 | rs2033939 | C15orf54 | Intergenic | 22,759 | 8.49E−14 |
| | | 7q21.3 | 96129071 | rs62470042 | C7orf76 | Intronic | 22,759 | 7.38E−13 |
| | | 6q15 | 91942761 | rs12660096 | MAP3K7 | Intergenic | 22,759 | 4.74E−10 |
| | Lateral orbitofrontal | 14q22.2 | 54769839 | rs6572946 | CDKN3 | Intergenic | 22,801 | 2.29E−10 |
| | Precentral | 15q14 | 39634222 | rs1080066 | C15orf54 | Intergenic | 22,699 | 5.84E−125 |
| | | 10q25.3 | 118648841 | rs3781566 | SHTN1 | Intronic | 22,699 | 4.68E−11 |
| Temporal | Superior temporal | 3q26.32 | 177296448 | rs13084960 | LINC00578 | ncRNA_intronic | 22,681 | 1.12E−11 |
| | Banksts | 14q23.1 | 59072226 | rs186347 | KIAA0586 | Intergenic | 22,727 | 1.15E−15 |
| | Fusiform | 14q23.1 | 59833172 | rs1547199 | DAAM1 | Intronic | 22,605 | 4.58E−10 |
| | | 1p33 | 47980916 | rs6658111 | FOXD2 | Intergenic | 22,605 | 7.78E−10 |
| | Transverse temporal | 2q23.2 | 150012936 | rs2046268 | LYPD6B | Intronic | 22,786 | 2.55E−12 |
| | Parahippocampal | 2q33.1 | 199809716 | rs966744 | SATB2 | Intergenic | 22,747 | 2.23E−10 |
| Parietal | Superior parietal | 15q14 | 39633904 | rs2033939 | C15orf54 | Intergenic | 22,723 | 4.28E−23 |
| | | 16q24.2 | 87225139 | rs4843227 | LOC101928708 | Intergenic | 22,723 | 1.16E−13 |
| | | 19p13.2 | 13109763 | rs68175985 | NFIX | Intronic | 21,777 | 3.27E−11 |
| | | 5q15 | 92866553 | rs62369942 | NR2F1-AS1 | ncRNA_intronic | 21,664 | 4.32E−10 |
| | Inferior parietal | 20q13.2 | 52448936 | rs6097618 | SUMO1P1 | Intergenic | 22,701 | 2.09E−17 |
| | | 12q14.3 | 65797096 | rs2336713 | MSRB3 | Intronic | 22,701 | 2.47E−13 |
| | | 3q13.11 | 104724634 | rs971551 | ALCAM | Intergenic | 22,701 | 2.34E−10 |
| | Supramarginal | 15q14 | 39632013 | rs71471500 | THBS1 | Intergenic | 22,645 | 9.71E−28 |
| | | 14q23.1 | 59627631 | rs2164950 | DAAM1 | Intergenic | 22,645 | 3.59E−20 |
| | | 3q24 | 147106319 | rs2279829 | ZIC4 | UTR3 | 22,645 | 5.36E−18 |
| | Postcentral | 15q14 | 39633904 | rs2033939 | THBS1 | Intergenic | 22,662 | 4.34E−133 |
| | | 3q24 | 147106319 | rs2279829 | ZIC4 | UTR3 | 22,662 | 2.54E−17 |
| | | 9q21.13 | 76144318 | rs67286026 | ANXA1 | Intergenic | 22,662 | 5.03E−11 |
| | | 2q36.3 | 226563259 | rs16866701 | NYAP2 | Intergenic | 22,545 | 5.69E−11 |
| | Precuneus | 14q23.1 | 59628609 | rs74826997 | DAAM1 | Intergenic | 22,803 | 4.85E−20 |
| | | 3q28 | 190663557 | rs35055419 | OSTN | Intergenic | 22,428 | 2.02E−10 |
| | | 2p22.2 | 37818236 | rs2215605 | CDC42EP3 | Intergenic | 22,803 | 3.43E−10 |
| | | 3q13.11 | 104713881 | rs12495603 | ALCAM | Intergenic | 22,803 | 9.71E−10 |
| Occipital | Lateral occipital | 14q23.1 | 59627631 | rs2164950 | DAAM1 | Intergenic | 22,799 | 6.89E−16 |
| | Lingual | 14q23.1 | 59625997 | rs73313052 | DAAM1 | Intergenic | 22,805 | 1.06E−20 |
| | | 6q22.32 | 127089401 | rs2223739 | RSPO3 | Intergenic | 22,805 | 1.75E−10 |
| | Cuneus | 14q23.1 | 59625997 | rs73313052 | DAAM1 | Intergenic | 22,799 | 4.59E−43 |
| | | 11p15.3 | 12072213 | rs11022131 | DKK3 | Intergenic | 22,799 | 5.96E−12 |
| | | 13q31.1 | 80192236 | rs9545156 | LINC01068 | Intergenic | 22,799 | 4.09E−10 |
| | Pericalcarine | 14q23.1 | 59628679 | rs76341705 | DAAM1 | Intergenic | 22,824 | 1.39E−29 |
| | | 13q31.1 | 80191873 | rs9545155 | LINC01068 | intergenic | 22,824 | 2.25E−13 |
| | | 11p14.1 | 30876113 | rs273594 | DCDC5 | Intergenic | 22,824 | 3.51E−13 |
| | | 1p13.2 | 113208039 | rs12046466 | CAPZA1 | Intronic | 22,824 | 2.36E−12 |
| | | 1p33 | 47980916 | rs6658111 | FOXD2 | Intergenic | 22,824 | 3.85E−11 |
| | | 11q22.3 | 104012656 | rs1681464 | PDGFD | Intronic | 22,824 | 7.51E−11 |
| | | 6q22.32 | 127096181 | rs9401907 | RSPO3 | Intergenic | 22,824 | 2.11E−10 |
| | | 7p21.1 | 18904400 | rs12700001 | HDAC9 | Intronic | 22,824 | 2.12E−10 |
| | | 5q12.1 | 60315823 | rs10939879 | NDUFAF2 | Intronic | 22,824 | 2.92E−10 |
| | Caudal anterior cingulate | 5q14.3 | 82852578 | rs309588 | VCAN | Intronic | 22,748 | 2.60E−10 |
| | Insula | 11q23.1 | 110949402 | rs321403 | C11orf53 | Intergenic | 22,543 | 9.58E−12 |
| | | 8q24.12 | 120596023 | rs10283100 | ENPP2 | Exonic | 21,481 | 8.29E−11 |

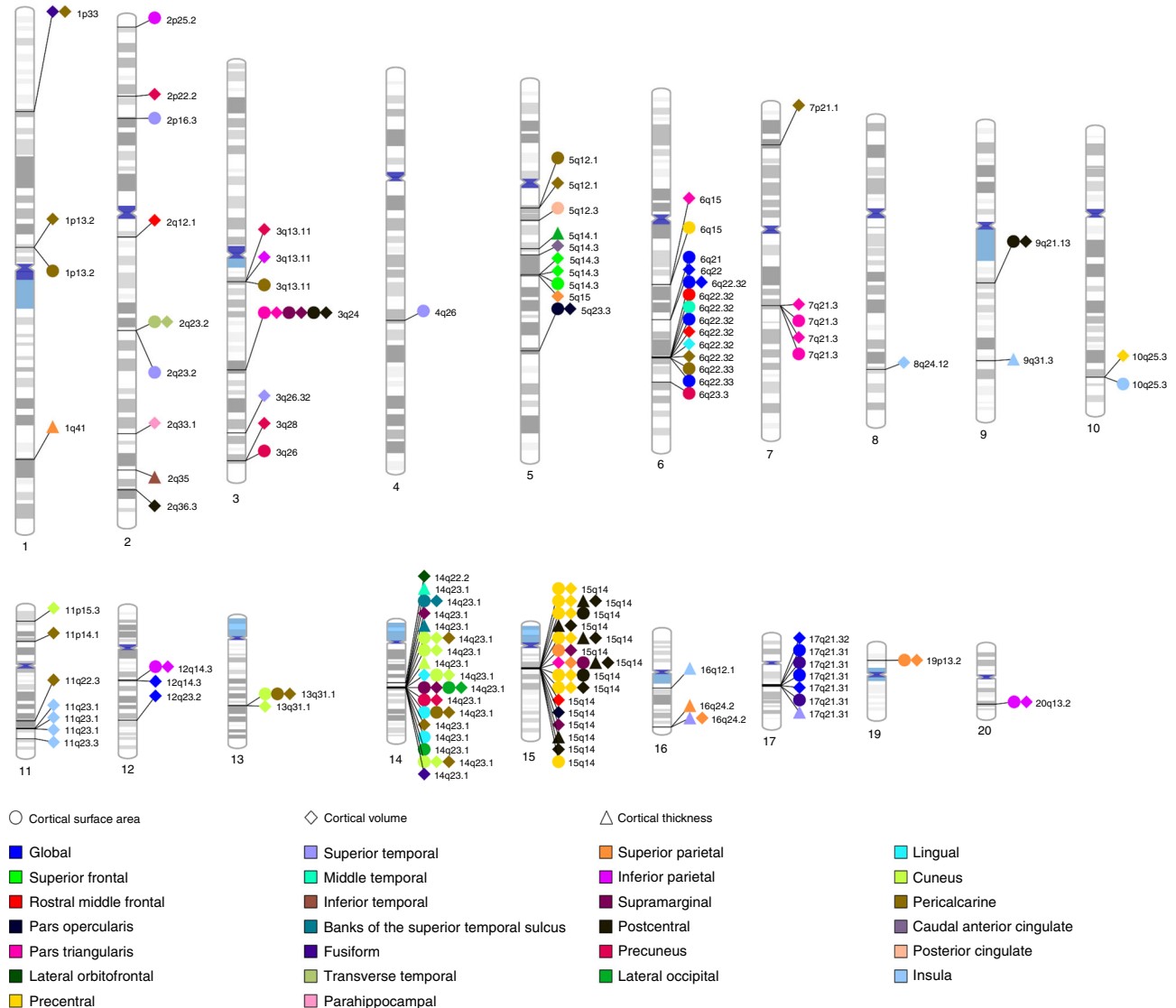

**Fig. 1 Chromosomal ideogram of genome-wide significant associations with measures of cortical structure.** Cortical surface areas, cortical volumes and cortical thickness. Each point represents the significantly associated variant, the colors correspond to the different cortical regions and the shape to different type of measurment ($p_{\text{Discovery}} < 1.09 \times 10^{-9}$).

Supplementary Data 22 and 23. For global CTh, we found enrichment for super-enhancers, introns and histone marks. Repressors and histone marks were enriched for global CSA, and introns, super-enhancers, and repressors for global CV. For regional CSA and CV the highest enrichment scores (>18) were observed for conserved regions.

**Genetic correlation**. We found high genetic correlation ($r_{\text{g}}$) between global CSA, and global CV ($r_{\text{g}} = 0.81$, $p_{\text{LDSR}} = 1.2 \times 10^{-186}$) and between global CTh and global CV ($r_{\text{g}} = 0.46$, $p_{\text{LDSR}} = 1.4 \times 10^{-14}$), but not between global CTh and global CSA ($r_{\text{g}} = -0.02$, $p_{\text{LDSR}} = 0.82$). Whereas the genetic correlation between CSA and CV was strong ($r_{\text{g}} > 0.7$) in most of the regions (Supplementary Data 24 and Supplementary Fig. 12), it was generally weak between CSA and CTh with $r_{\text{g}} < 0.3$, and ranged from 0.09 to 0.69 between CTh and CV. The postcentral and lingual cortices were the two regions with the highest genetic correlations between both CTh and CV, as well as CTh and CSA.

Genetic correlation across the various brain regions for CTh (Supplementary Fig. 13, Supplementary Data 25), CSA (Supplementary Fig. 14, Supplementary Data 26), and CV

(Supplementary Fig. 15, Supplementary Data 27) showed a greater number of correlated regions for CTh and greater inter-regional variation for CSA and CV. Supplementary Data 28–30 and Supplementary Figs. 16–18 show genome-wide genetic correlations between the cortical measures and anthropometric, neurological and psychiatric, and cerebral structural traits.

## Discussion

In our genome-wide association study of up to 22,824 individuals for MRI determined cortical measures of global and regional thickness, surface area, and volume, we identified 160 genome-wide significant associations across 19 chromosomes. Heritability was generally higher for cortical surface area and volume than for thickness, suggesting a greater susceptibility of cortical thickness to environmental influences. We observed strong genetic correlations between surface area and volume, but weak genetic correlation between surface area and thickness. We identified the largest number of novel genetic associations with cortical volumes, perhaps due to our larger sample size for this phenotype, which was assessed in all 21 discovery samples.

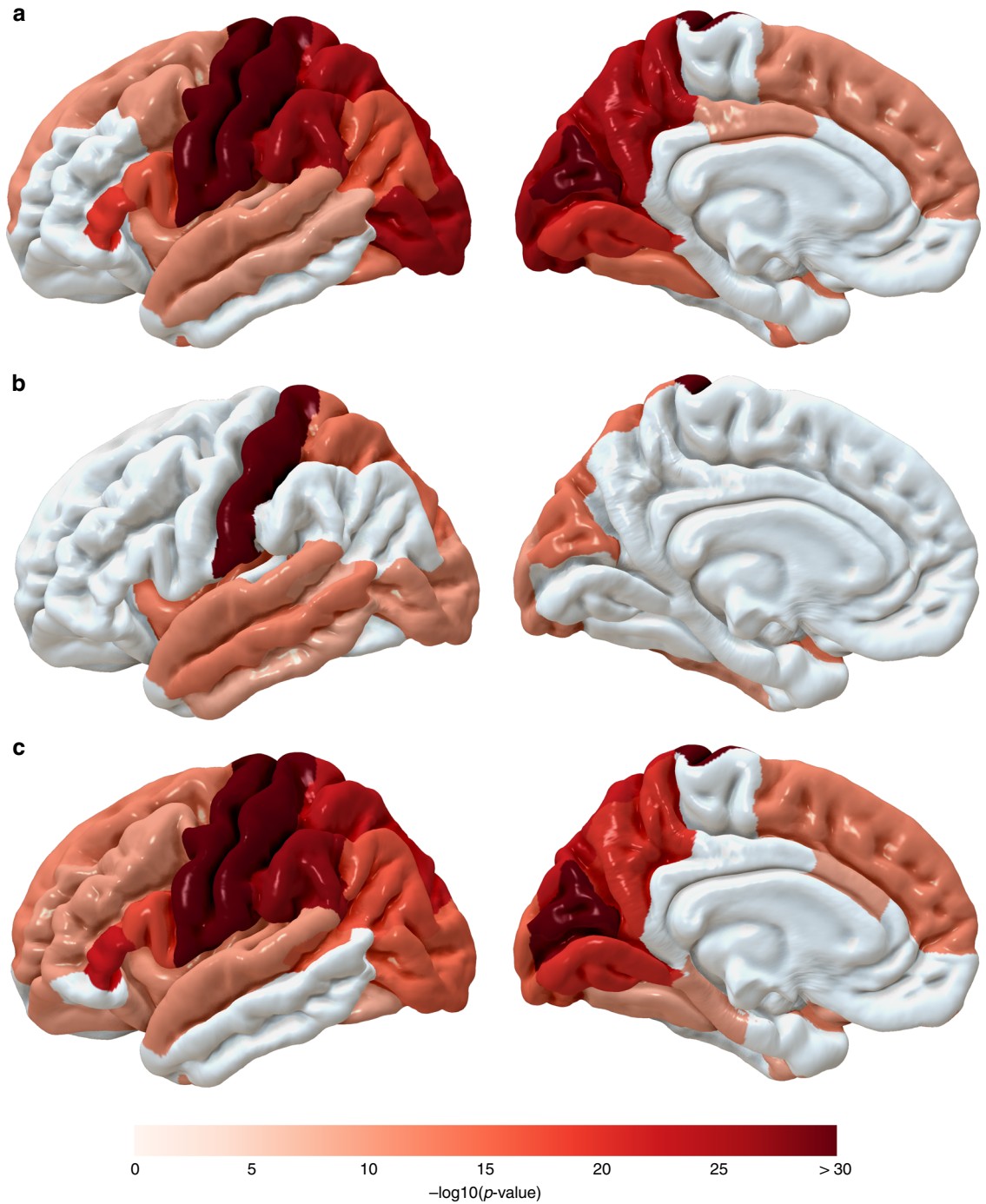

**Fig. 2 Lowest discovery meta-analysis *p*-value of CSA, CTh, and CV in each cortical region. a** Lowest $p_{Discovery}$ of CSA, **b** lowest $p_{Discovery}$ of CTh, **c** lowest $p_{Discovery}$ of CV.

It is beyond the scope of our study to discuss each of the 160 associations identified. A large number of the corresponding genes are involved in pathways that regulate morphogenesis of neurons, neuronal cell differentiation, and cell growth, as well as cell migration and organogenesis during embryonic development. At a molecular level, the wnt/β-catenin, TGF-β, and sonic hedgehog pathways are strongly implicated. Gene-set-enrichment analyses revealed biological processes related to brain morphology and neuronal development.

Broad patterns emerged showing that genes determining cortical structure are also often implicated in development of the cerebellum and brainstem (*KIAA0586, ZIC4, ENPP2*) as well as

the neural tube (one carbon metabolism genes *DHFR* and *MSRBB3*, the latter also associated with hippocampal volumes[25]). These genes determine development of not only neurons but also astroglia (*THBS1*) and microglia (*SALL1*). They determine susceptibility or resistance to a range of insults: inflammatory, vascular (*THBS1, ANXA1, ARRDC3-AS1*[26]) and neurodegenerative (*C15orf53, ZIC4, ANXA1*), and have been associated with pediatric and adult psychiatric conditions (*THBS1*).

There is a wealth of information in the supplementary tables that can be mined for a better understanding of brain development, connectivity, function and pathology. We highlight this potential by discussing in additional detail, the possible

significance of 6 illustrative loci, 5 of which, at 15q14, 14q23.1, 6q22.32, 17q21.31, and 3q24, associate with multiple brain regions at low *p*-values, while the locus at 8q24.12 identifies a plausible exonic variant.

The Chr15q14 locus was associated with cortical thickness, surface area, and volumes in the postcentral gyrus as well as with surface area or volume across six other regions in the frontal and parietal lobes. Lead SNPs at this locus were either intergenic between *C15orf53* and *C15orf54*, or intergenic between *C15orf54* and *THBS1* (Thrombospondin-1). *C15orf53* has been associated with an autosomal recessive form of spastic paraplegia showing intellectual disability and thinning of the corpus callosum (hereditary spastic paraparesis 11, or Nakamura Osame syndrome). Variants of *THBS1* were reported to be related to autism[27] and schizophrenia[28]. The protein product of *THBS1* is involved in astrocyte induced synaptogenesis[29], and regulates chain migration of interneuron precursors migrating in the postnatal radial migration stream to the olfactory bulb[30]. Moreover, *THBS1* is an activator of TGFβ signaling, and an inhibitor of pro-angiogenic nitric oxide signaling, which plays a role in several cancers and immune-inflammatory conditions.

Variants at Chr14q23.1 were associated with cortical surface area and volume of all regions in the occipital lobe, as well as with thickness, surface area, and volume of the middle temporal

cortex, banks of the superior temporal sulcus, fusiform, supramarginal and precuneus regions, areas associated with discrimination and recognition of language or visual form. These variants are either intergenic between *KIAA0586*, the product of which is a conserved centrosomal protein essential for ciliogenesis, sonic hedgehog signaling and intracellular organization, and *DACT1*, the product of which is a target for *SIRT1* and acts on the wnt/β-catenin pathway. *KIAA0586* has been associated with Joubert syndrome, another condition associated with abnormal cerebellar development. Other variants are intergenic between *DACT1* and *DAAM1* or intronic in *DAAM1*. *DAAM1* has been associated with occipital lobe volume in a previous GWAS[31].

Locus 6q22.32 contains various SNPs associated with cortical surface area and volume globally, and also within some frontal, temporal and occipital regions. The SNPs are intergenic between *RSPO3* and *CENPW*. *RSPO3* and *CENPW* have been previously associated with intracranial[32,33] and occipital lobe volumes[31]. *RSPO3* is an activator of the canonical Wnt signaling pathway and a regulator of angiogenesis.

Chr17q21.31 variants were associated with global cortical surface area and volume and with regions in temporal lobe. These variants are intronic in the genes *PLEKHM1*, *CRHR1*, *NSF*, and *WNT3*. In previous GWAS analyses, these genes have been associated with general cognitive function[34] and neuroticism[35]. *CRHR1*, *NSF*, and *WNT3* were additionally associated with Parkinson's disease[36] and intracranial volume[32,33,37]. The *NSF* gene also plays a role in Neuronal Intranuclear Inclusion Disease[38] and *CRHR1* is involved in anxiety and depressive disorders[39]. This chromosomal region also contains the *MAPT* gene, which plays a role in Alzheimer's disease, Parkinson's disease, and frontotemporal dementia[40,41].

The protein product of the gene *ZIC4* is a C2H2 zinc finger transcription factor that has an intraneuronal, non-synaptic expression and auto-antibodies to this protein have been associated with subacute sensory neuronopathy, limbic encephalitis, and seizures in patients with breast, small cell lung or ovarian cancers. *ZIC4* null mice have abnormal development of the visual pathway[42] and heterozygous deletion of the gene has also been associated with a congenital cerebellar (Dandy-Walker) malformation[43], thus implicating it widely in brain development as well as in neurodegeneration. *C2H2ZF* transcription factors are the most widely expressed transcription factors in eukaryotes and show associations with responses to abiotic (environmental) stressors. Another transcription factor, *FOXC1*, also associated with Dandy-Walker syndrome has been previously shown to be associated with risk of all types of ischemic stroke and with stroke

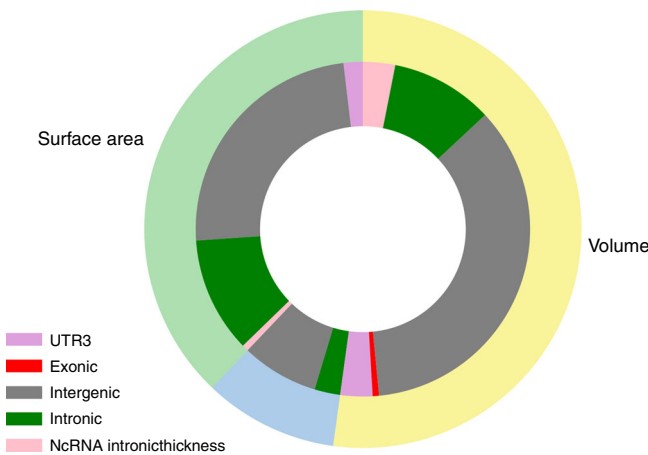

**Fig. 3 Functional annotation categories for global and regional CTh, CSA, and CV.** Proportion of functional annotation categories for global and regional cortical thickness (blue), surface area (light green), and volume (yellow) assigned by ANNOVAR.

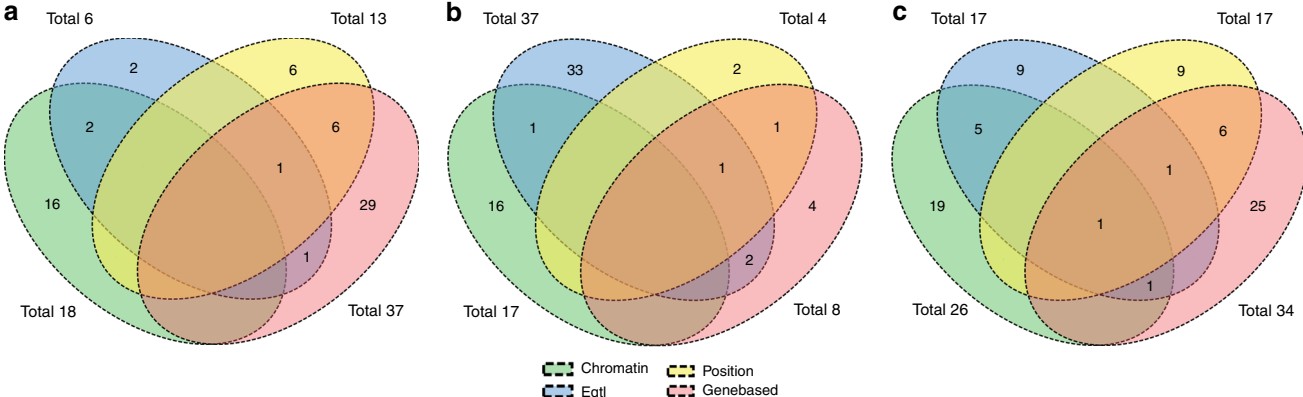

**Fig. 4 Number of overlapping genes between gene mapping methods.** Number of overlapping genes between FUMA eQTL mapping, FUMA chromatin interaction mapping, ANNOVAR chromosome positional mapping, and MAGMA gene-based analysis for all cortical regions combined for cortical surface area (**a**), thickness (**b**) and volume (**c**).

severity. Thus, *ZIC4* might be a biological target worth pursuing to ameliorate neurodegenerative disorders.

We found an exonic SNP within the gene *ENPP2* (Autotaxin) at 8q24.12 to be associated with insular cortical volume. This gene is differentially expressed in the frontal cortex of Alzheimer patients[44] and in mouse models of Alzheimer disease, such as the senescence-accelerated mouse prone 8 strain (SAMP8) mouse. Autotaxin is a dual-function ectoenzyme, which is the primary source of the signaling lipid, lysophosphatidic acid. Besides Alzheimer disease, changes in autotaxin/lysophosphatidic acid signaling have also been shown in diverse brain-related conditions, such as intractable pain, pruritus, glioblastoma, multiple sclerosis, and schizophrenia. In the SAMP8 mouse, improvements in cognition noted after administration of LW-AFC, a putative Alzheimer remedy derived from the traditional Chinese medicinal prescription 'Liuwei Dihuang' decoction, are correlated with restored expression of four genes in the hippocampus, one of which is *ENPP2*.

Among the other genetic regions identified, many have been linked to neurological and psychiatric disorders, cognitive functioning, cortical development, and cerebral structure (detailed listing in Supplementary Data 31).

Heritability estimates are, as expected, generally higher in the family-based Austrian Stroke Prevention-Family study (ASPS-Fam) than in the Rotterdam Study (RS) for CTh (average $h^2_{ASPS-Fam} = 0.52$; $h^2_{RS} = 0.26$), CSA (0.62 and 0.30) and CV (0.57 and 0.23). This discrepancy is explained by the different heritability estimation methods: pedigree-based heritability in ASPS-Fam versus heritability based on common SNPs that are in LD with causal variants[45] in RS.

Average heritability over regions is also higher for surface area and volume, than for thickness. The observed greater heritability of CSA compared to CTh is consistent with the previously articulated hypothesis, albeit based on much smaller numbers, that CSA is developmentally determined to a greater extent with smaller subsequent decline after young adulthood, whereas CTh changes over the lifespan as aging, neurodegeneration and vascular injuries accrue[1,3]. It is also interesting that brain regions more susceptible to early amyloid deposition (e.g., superior temporal cortex and precuneus) have a higher heritability.

We found no or weak genetic correlation between CTh and CSA, globally and regionally, and no common lead SNPs, which indicates that these two morphological measures are genetically independent, a finding consistent with prior reports[15,16]. In contrast, we found strong genetic correlation between CSA and CV and identified common lead SNPs for CSA and CV globally, and in 12 cortical regions. Similar findings have been reported in a previous publication[16]. The genetic correlation between CTh and CV ranged between 0.09 and 0.77, implying a common genetic background in some regions (such as the primary sensory postcentral and lingual cortices), but not in others. For CTh, we observed genetic correlations between multiple regions within each of the lobes, whereas for CSA and CV, we found genetic correlations mainly between different regions of the occipital lobe. Chen et al.[46] have also reported strong genetic correlation for CSA within the occipital lobe. There were also a few genetic correlations observed for regions from different lobes, suggesting similarities in cortical development transcended traditional lobar boundaries.

A limitation of our study is the heterogeneity of the MR phenotypes between cohorts due to different scanners, field strengths, MR protocols and MRI analysis software. This heterogeneity as well as the different age ranges in the participating cohorts may have caused different effects over the cohorts. We nevertheless combined the data of the individual cohorts to maximize the sample size as it has been done in previous

CHARGE GWAS analyses[31–33]. To account for the heterogeneity we used a sample-size weighted meta-analysis that does not provide overall effect estimates. This method has lower power to detect associations compared to inverse-variance weighted meta-analysis and we therefore may have found less associations. Our inability to replicate 8 of the 76 genome-wide significant findings for CTh and CSA could be caused by false-positive results but may also be explained by insufficient power due to a too small sample size. Moreover, our sample comprises of mainly European ancestry, limiting the generalizability to other ethnicities. Strengths of our study are the population-based design, the large age range of our sample (20–100 years), use of three cortical measures as phenotypes of cortical morphometry, and the replication of our CTh and CSA findings in a large and independent cohort. In conclusion, we identified patterns of heritability and genetic associations with various global and regional cortical measures, as well as overlap of MRI cortical measures with genetic traits and diseases that provide new insights into cortical development, morphology, and possible mechanisms of disease susceptibility.

## Methods

**Study population.** The sample of this study consist of up to 22,824 participants from 20 population-based cohort studies collaborating in the Cohorts of Heart and Aging Research in Genomic Epidemiology (CHARGE) consortium and the UK Biobank (UKBB). All the individuals were stroke- and dementia free, aged between 20 and 100 years, and of European ancestry, except for ARIC AA with African ancestry. Supplementary Data 32 provides population characteristics of each cohort and the Supplementary Methods provide a short description of each study. Each study secured approval from institutional review boards or equivalent organizations, and all participants provided written informed consent. Our results were replicated using summary GWAS findings of 22,635 individuals from the ENIGMA consortium.

**Genotyping and imputation.** Genotyping was conducted using various commercially available genotyping arrays across the study cohorts. Prior to imputation, extensive quality control was performed in each cohort. Genotype data were imputed to the 1000 Genomes reference panel (mainly phase 1, version 3) using validated software. Details on genotyping, quality control and imputation can be found in Supplementary Data 33.

**Phenotype definition.** This study investigated CTh, CSA, and CV globally in the whole cortex and in 34 cortical regions. Global and regional CTh was defined as the mean thickness of the left and the right hemisphere in millimeter (mm). Global CSA was defined as the total surface area of the left and the right hemisphere in mm², while regional CSA was defined as the mean surface area of the left and the right hemisphere in mm². Global and regional CV was defined as the mean volume of the left and the right hemisphere in mm³. The 34 cortical regions are listed in the Supplementary Methods. High resolution brain magnetic resonance imaging (MRI) data was obtained in each cohort using a range of MRI scanners, field strengths and protocols. CTh, CSA, and CV were generated using the Freesurfer software package[47] in all cohorts except for FHSucd, where an in-house segmentation method was used. MRI protocols of each cohort can be found in Supplementary Data 35 and descriptive statistics of CTh, CSA, and CV can be found in Supplementary Data 36–38.

**Genome-wide association analysis.** Based on a predefined analysis plan, each study fitted linear regression models to determine the association between global and regional CTh, CSA, and CV and allele dosages of SNPs. Additive genetic effects were assumed and the models were adjusted for sex, age, age², and if needed for study site and for principal components to correct for population stratification. Cohorts including related individuals calculated linear mixed models to account for family structure. Details on association software and covariates for each cohort are shown in Supplementary Data 33. Models investigating regional CTh, CSA, and CV were additionally adjusted for global CTh, global CSA and global CV, respectively. Quality control of the summary statistics shared by each cohort was performed using EasyQC[48]. Genetic variants with a minor allele frequency (MAF) <0.05, low imputation quality ($R^2 < 0.4$), and which were available in less than 10,000 individuals were removed from the analyses. Details on quality control are provided in the Supplementary Methods.

We then used METAL[49] to perform meta-analyses using the *z*-scores method, based on *p*-values, sample size, and direction of effect, with genomic control correction. To estimate the number of independent tests for the p-value threshold correction, we used a non-parametric permutation testing procedure[50–53] in the

combined Rotterdam Study cohort ($N = 4442$) and UK Biobank ($N = 8213$). First, we generated a random independent variable, to insure that there is no true relationship between brain measurements and this variable. Second, we ran linear regression analyses between this variable and all brain measurements one-by-one in each of the cohorts separately (104 regressions in total per cohort). Third, we saved the minimum $p$-value obtained from those 104 regressions. Then, as suggested in literature[54], we repeated this procedure 10.000 times. Therefore, at the end we had 10.000 minimum $p$-values per cohort. The minimum $p$-value distribution follows a Beta distribution Beta($m,n$), where $m = 1$ and $n$ is the degree of freedom, which represents the number of independent tests in case of permutation testing. Using python statistical library we fitted the Beta function with the saved minimum $p$-values, and found n for Rotterdam Study and UK Biobank identically equal to 46. Based on the permutation test results, the genome-wide significance threshold was set a priori at $1.09 \times 10^{-9}$ ($= 5 \times 10^{-8}/46$). We used the clumping function in PLINK[55] (linkage disequilibrium (LD) threshold: 0.2, distance: 300 kb) to identify the most significant SNP in each LD block. We used LDSR to calculate genomic inflation factors ($\lambda_{GC}$), LDSR intercepts and LDSR ratios for each meta-analysis. The LDSR intercept was estimated to differentiate between inflation due to a polygenic signal and inflation due to population stratification[56]. The LDSR ratio represents the amount of inflation that is due to other causes than polygenicity such as population stratification or cryptic relatedness.

For replication of our genome-wide significant CTh and CSA associations, we used GWAS meta-analysis results from the ENIGMA consortium[22] for all SNPs that were associated at a $p$-value $<5 \times 10^{-8}$ and performed a pooled meta-analysis. The $p$-value threshold for replication was set to $3.1 \times 10^{-4}$ ($= 0.05/160$: nominal significance threshold divided by total number of lead SNPs). CV was not available in the ENIGMA results. PRS analysis was performed for 7800 out of sample subjects (not included in the current GWAS) from UK Biobank cohort using the PRSice-2 software[57] with standard settings. The significance threshold for the association between the PRS and the phenotype was set to $4.76 \times 10^{-3}$ ($= 0.05/105$: nominal significance divided by number GWAS phenotypes). The NHGRI-EBI Catalog of published GWAS[58] was searched for previous SNP-trait associations at a $p$-value of $5 \times 10^{-8}$ of lead SNPs. Regional association plots were generated with LocusZoom[59], and the chromosomal ideogram with PHENOGRAM (http://visualization.ritchielab.org/phenograms/plot).

Annotation of genome-wide significant variants was performed using the ANNOVAR software package[60] and the FUMA web application[61]. FUMA eQTL mapping uses information from three data repositories (GTEx, Blood eQTL browser, and BIOS QTL browser) and maps SNPs to genes based on a significant eQTL association. We used a false discovery rate threshold (FDR) of 0.05 divided by number of tests (46) to define significant eQTL associations. Gene-based analyses, to combine the effects of SNPs assigned to a gene, and gene set analyses, to find out if genes assigned to significant SNPs were involved in biological pathways, were performed using MAGMA[62] as implemented in FUMA. The significance threshold was set to $5.87 \times 10^{-8}$ ($= 0.05/18522*46$: FDR threshold divided by number of genes and independent tests) for gene-based analyses and to $1.02 \times 10^{-7}$ ($= 0.05/10651$: FDR threshold divided by the number of gene sets) for the gene set analyses. Additionally, FUMA was used to investigate a significant chromatin interaction between a genomic region in a risk locus and promoter regions of genes (250 bp upstream and 500 bp downstream of a TSS). We used an FDR of $1 \times 10^{-6}$ to define significant interactions.

We investigated cis ($<1$ Mb) and trans ($>1$ MB or on a different chromosome) expression quantitative trait loci (eQTL) for genome-wide significant SNPs in 724 post-mortem brains from ROSMAP[63,64] stored in the AMP-AD database. The samples were collected from the gray matter of the dorsolateral prefrontal cortex. The significance threshold was set to 0.001 ($= 0.05/46$: FDR threshold divided by the number of independent tests). For additional pathway analyses of genes that were mapped to independent lead SNPs by FUMA, we searched the InnateDB database[65]. The STRING database[66] was used for visualizing protein–protein interactions. Only those protein subnetworks with five or more nodes are shown.

**Heritability**. Additive genetic heritability ($h^2$) of CTh, CSA, and CV was estimated in two studies: the Austrian Stroke Prevention Family Study (ASPS-Fam; $n = 365$) and the Rotterdam Study (RS, $n = 4472$). In the population-based family study ASPS-Fam, the ratio of the genotypic variance to the phenotypic variance was calculated using variance components models in SOLAR[67]. In case of non-normality, phenotype data were inverse-normal transformed. In RS, SNP-based heritability was computed with GCTA[68]. These heritability analyses were adjusted for age and sex.

Heritability and partitioned heritability based on GWAS summary statistics was calculated from GWAS summary statistics using LDSR) implemented in the LDSC tool (https://github.com/bulik/ldsc). Partitioned heritability analysis splits genome-wide SNP heritability into 53 functional annotation classes (e.g., coding, 3′UTR, promoter, transcription factor binding sites, conserved regions etc.) and additionally to 10 cell-type specific classes (e.g., central nervous system, cardiovascular, liver, skeletal muscle, etc.) as defined by Finucane et al.[69] to estimate their contributions to heritability. The significance threshold was set to $2.05 \times 10^{-5}$ ($= 0.05/53*46$: nominal significance divided by number of functional annotation classes and number of independent tests) for heritability partitioned on

functional annotation classes and $2.05 < 10^{-6}$ ($= 0.05/53*10*46$: nominal significance divided by number of functional annotation classes, number of cell types and number of independent tests) for heritability partitioned on annotation classes and cell types.

**Genetic correlation**. LDSR genetic correlation[70] between CTh, CSA, and CV was estimated globally and within each cortical region. The significance threshold was set to $7.35 \times 10^{-4}$ (nominal threshold (0.05) divided by number of regions (34) and by number of correlations (CSA and CV, CSA and CTh). Genetic correlation was also estimated between all 34 cortical regions for CTh, CSA, and CV, with the significance threshold set to $8.91 \times 10^{-5}$ (nominal threshold (0.05) divided by number of regions (34) times the number of regions $-1$ (33) divided by 2 (half of the matrix). Additionally, the amount of genetic correlation was quantified between CTh, CSA, and CV and physical traits (height, body mass index), neurological and psychiatric diseases (e.g., Alzheimer's disease, Parkinson's disease), cognitive traits and MRI volumes ($p$-value threshold (0.05/46/number of GWAS traits). As recommended by the LDSC tool developers, only HapMap3 variants were included in these analyses, as these tend to be well-imputed across cohorts.

**Reporting summary**. Further information on research design is available in the Nature Research Reporting Summary linked to this article.

## Data availability

The genome-wide summary statistics that support the findings of this study are available via the CHARGE Summary Results portal at the NCBI dbGaP website https://www.omicsdi.org/dataset/dbgap/phs000930 upon publication, or from the corresponding authors R.S. and S.S. upon reasonable request. The summary statistics may be used for all scientific purposes except for the study of potentially sensitive and potentially stereotyping phenotypes such as intelligence and addiction, since this is proscribed by the consent terms for the NHLBI cohorts. Individual level data or study-specific summary results are only available through controlled access. Data for the Framingham Study are available through dbGaP, where qualified researchers can apply for authorization to access (https://www.ncbi.nlm.nih.gov/projects/gap/cgi-bin/study.cgi?study_id=phs000007.v30.p11). Individual level data for the ARIC and CHS studies are also available through dbGaP. Data of European and Australian cohorts are available upon request, in keeping with data sharing guidelines in the EU General Data Protection Regulation. Data from UK Biobank can be accessed at http://www.ukbiobank.ac.uk and for the ENIGMA consortium from medlandse@gmail.com. Individual level data for VETSA is not available due to consent restrictions.

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

## Acknowledgements

We provide all investigator and study-specific acknowledgements in Supplementary Note 1.

## Author contributions

Drafting of the manuscript: E.H., G.V.R., R.S., and S.S.; Genotype and phenotype data acquisition: H.S., N.A., P.R.S., M.J.W., C.E.F., P.S.S., K.A.M, N.J.A., J.B.K., O.A.A, A.M.D, M.C.N, U.V., R.S., S.M., B.M., J.J., J.T., G.B.P., H.B., A.T., D.A., RB, J.T.B., O.L.L., C.T., and P.A.; Imaging, genetic and bioinformatics analysis: E.H., G.V.R., H.H.H.A., M.J.K., Y.S., R.X., J.C.B., S.A., S.L., S.J.v.d.L., Q.Y., C.L.S., H.L., J.J.H., H.Z., M.L., M.S., N.J.A., M.W.V., A.T., K.W., M.B., A.M., N.A.G., M.L., P.R.S., X.J., O.C., A.S.B, L.P., St. S., P.M., C.dC., S.K., L.L., M.B., W.W., F.B., A.V.W., M.H., J.J., and F.C. Cohort PIs: P.S.S., W.S.K., J.A.W., A.V., C.M.v. D, H.J.G., W.T.L.Jr, M.F., T.P., S.D., M.A.I., H.S., R.S., S.S., J.I.R., B.M.P., I.J.D., M.L., A.H., A.G.U., W.J.N., Z.P., M.S.P., T.H.M., J.T., C.E.F., M.J.L., and C.T.; ENIGMA Study design: K.L.G., N.J., J.N.P., L.C.-C., J.B., D.P.H., P.A.L., F.P., J.L.S., P.M.T., and S.E.M.; Critical revision of the manuscript: all authors.

## Competing interests

Dr. Dale is a founder of and holds equity in CorTechs Labs, Inc, and serves on its Scientific Advisory Board. He is a member of the Scientific Advisory Board of Human Longevity, Inc. and receives funding through research agreements with General Electric Healthcare and Medtronic, Inc. The terms of these arrangements have been reviewed and approved by UCSD in accordance with its conflict of interest policies. W. Niessen is co-founder and shareholder of Quantib BV. H. Brodaty is an advisory board member of Nutricia. The other authors declare no competing interests.

## Additional information

Edith Hofer[1,2,229], Gennady V. Roshchupkin[3,4,5,229], Hieab H. H. Adams[3,5,229], Maria J. Knol[5], Honghuang Lin[6], Shuo Li[7], Habil Zare[8,9], Shahzad Ahmad[5], Nicola J. Armstrong[10], Claudia L. Satizabal[8,11], Manon Bernard[12], Joshua C. Bis[13], Nathan A. Gillespie[14,15], Michelle Luciano[16,17], Aniket Mishra[18], Markus Scholz[19,20], Alexander Teumer[21], Rui Xia[22], Xueqiu Jian[22], Thomas H. Mosley[23], Yasaman Saba[24], Lukas Pirpamer[1], Stephan Seiler[25,26], James T. Becker[27], Owen Carmichael[28], Jerome I. Rotter[29], Bruce M. Psaty[13], Oscar L. Lopez[27], Najaf Amin[5], Sven J. van der Lee[5], Qiong Yang[7], Jayandra J. Himali[7], Pauline Maillard[25,26], Alexa S. Beiser[7,11], Charles DeCarli[25,26], Sherif Karama[30], Lindsay Lewis[30], Mat Harris[16,31,32,33], Mark E. Bastin[16,31,32,33], Ian J. Deary[16,17], A. Veronica Witte[34,35], Frauke Beyer[34,35], Markus Loeffler[19,20], Karen A. Mather[36,37], Peter R. Schofield[37,38], Anbupalam Thalamuthu[36], John B. Kwok[38,39], Margaret J. Wright[40,41], David Ames[42,43], Julian Trollor[36,44], Jiyang Jiang[36], Henry Brodaty[45,36], Wei Wen[36], Meike W. Vernooij[3,5], Albert Hofman[46,5], André G. Uitterlinden[5], Wiro J. Niessen[47,3], Katharina Wittfeld[48,49], Robin Bülow[50], Uwe Völker[51], Zdenka Pausova[12,52], G. Bruce Pike[53], Sophie Maingault[54], Fabrice Crivello[54], Christophe Tzourio[18,55], Philippe Amouyel[56,57,58], Bernard Mazoyer[54], Michael C. Neale[14], Carol E. Franz[59], Michael J. Lyons[60], Matthew S. Panizzon[59], Ole A. Andreassen[61], Anders M. Dale[62], Mark Logue[7,63,64], Katrina L. Grasby[65], Neda Jahanshad[66], Jodie N. Painter[65], Lucía Colodro-Conde[65], Janita Bralten[67,68], Derek P. Hibar[66,69], Penelope A. Lind[65], Fabrizio Pizzagalli[66], Jason L. Stein[70], Paul M. Thompson[66], Sarah E. Medland[65], ENIGMA consortium*, Perminder S. Sachdev[36,71], William S. Kremen[59], Joanna M. Wardlaw[16,31,32,33], Arno Villringer[34,72], Cornelia M. van Duijn[5,73], Hans J. Grabe[48,49], William T. Longstreth Jr[74], Myriam Fornage[22], Tomas Paus[75,76], Stephanie Debette[11,18,77], M. Arfan Ikram[3,5,78], Helena Schmidt[24], Reinhold Schmidt[1,230]✉ & Sudha Seshadri[8,11,230]✉

[1]Clinical Division of Neurogeriatrics, Department of Neurology, Medical University of Graz, Graz, Austria. [2]Institute for Medical Informatics, Statistics and Documentation, Medical University of Graz, Graz, Austria. [3]Department of Radiology and Nuclear Medicine, Erasmus MC, Rotterdam, The Netherlands. [4]Department of Medical Informatics, Erasmus MC, Rotterdam, The Netherlands. [5]Department of Epidemiology, Erasmus MC, Rotterdam, The Netherlands. [6]Section of Computational Biomedicine, Department of Medicine, Boston University School of

Medicine, Boston, MA, USA. [7]Department of Biostatistics, Boston University School of Public Health, Boston, MA, USA. [8]Glenn Biggs Institute for Alzheimer's and Neurodegenerative Diseases, UT Health San Antonio, San Antonio, USA. [9]Department of Cell Systems & Anatomy, The University of Texas Health Science Center, San Antonio, TX, USA. [10]Mathematics and Statistics, Murdoch University, Perth, Australia. [11]Department of Neurology, Boston University School of Medicine, Boston, MA, USA. [12]Hospital for Sick Children, Toronto, ON, Canada. [13]Cardiovascular Health Research Unit, Department of Medicine, Epidemiology and Health Services, University of Washington, Seattle, WA, USA. [14]Virginia Institute for Psychiatric and Behavior Genetics, Virginia Commonwealth University, Richmond, VA, USA. [15]QIMR Berghofer Medical Research Institute, Herston, QLD, Australia. [16]Centre for Cognitive Epidemiology and Cognitive Ageing, University of Edinburgh, Edinburgh, UK. [17]Department of Psychology, University of Edinburgh, Edinburgh, UK. [18]University of Bordeaux, Bordeaux Population Health Research Center, INSERM UMR 1219, Bordeaux, France. [19]Institute for Medical Informatics, Statistics and Epidemiology, University of Leipzig, Leipzig, Germany. [20]LIFE Research Center for Civilization Diseases, University of Leipzig, Leipzig, Germany. [21]Institute for Community Medicine, University Medicine Greifswald, Greifswald, Germany. [22]Institute of Molecular Medicine and Human Genetics Center, University of Texas Health Science Center at Houston, Houston, TX, USA. [23]Department of Medicine, University of Mississippi Medical Center, Jackson, MS, USA. [24]Gottfried Schatz Research Center for Cell Signaling, Metabolism and Aging, Medical University of Graz, Graz, Austria. [25]Imaging of Dementia and Aging (IDeA) Laboratory, Department of Neurology, University of California-Davis, Davis, CA, USA. [26]Department of Neurology and Center for Neuroscience, University of California at Davis, Sacramento, CA, USA. [27]Departments of Psychiatry, Neurology, and Psychology, University of Pittsburgh, Pittsburgh, PA, USA. [28]Pennington Biomedical Research Center, Baton Rouge, LA, USA. [29]Institute for Translational Genomics and Population Sciences, Los Angeles Biomedical Research Institute and Pediatrics at Harbor-UCLA Medical Center, Torrance, CA, USA. [30]McGill University, Montreal Neurological Institute, Montreal, QC, Canada. [31]Centre for Clinical Brain Sciences, University of Edinburgh, Edinburgh, UK. [32]Brain Research Imaging Centre, University of Edinburgh, Edinburgh, UK. [33]Scottish Imaging Network, A Platform for Scientific Excellence (SINAPSE) Collaboration, Department of Neuroimaging Sciences, The University of Edinburgh, Edinburgh, UK. [34]Department of Neurology, Max Planck Institute for Human Cognitive and Brain Sciences, Leipzig, Germany. [35]Faculty of Medicine, CRC 1052 Obesity Mechanisms, University of Leipzig, Leipzig, Germany. [36]Centre for Healthy Brain Ageing, School of Psychiatry, University of New South Wales, Sydney, Australia. [37]Neuroscience Research Australia, Sydney, Australia. [38]School of Medical Sciences, University of New South Wales, Sydney, Australia. [39]Brain and Mind Centre - The University of Sydney, Camperdown, NSW, Australia. [40]Queensland Brain Institute, The University of Queensland, St Lucia, QLD, Australia. [41]Centre for Advanced Imaging, The University of Queensland, St Lucia, QLD, Australia. [42]National Ageing Research Institute, Royal Melbourne Hospital, Parkvill, VIC, Australia. [43]Academic Unit for Psychiatry of Old Age, University of Melbourne, St George's Hospital, Kew, VIC, Australia. [44]Department of Developmental Disability Neuropsychiatry, School of Psychiatry, University of New South Wales, Sydney, NSW, Australia. [45]Dementia Centre for Research Collaboration, University of New South Wales, Sydney, NSW, Australia. [46]Department of Epidemiology, Harvard T.H. Chan School of Public Health, Boston, MA, USA. [47]Imaging Physics, Faculty of Applied Sciences, Delft University of Technology, Delft, The Netherlands. [48]German Center for Neurodegenerative Diseases (DZNE), Site Rostock/Greifswald, Greifswald, Germany. [49]Department of Psychiatry and Psychotherapy, University Medicine Greifswald, Greifswald, Germany. [50]Institute for Diagnostic Radiology and Neuroradiology, University Medicine Greifswald, Greifswald, Germany. [51]Interfaculty Institute for Genetics and Functional Genomics, University Medicine Greifswald, Greifswald, Germany. [52]Departments of Physiology and Nutritional Sciences, University of Toronto, Toronto, ON, Canada. [53]Departments of Radiology and Clinial Neurosciences, University of Calgary, Calgary, AB, Canada. [54]Institut des Maladies Neurodégénratives UMR5293, CEA, CNRS, University of Bordeaux, Bordeaux, France. [55]Pole de santé publique, Centre Hospitalier Universitaire de Bordeaux, Bordeaux, France. [56]Centre Hospitalier Universitaire de Bordeaux, France; Inserm U1167, Lille, France. [57]Department of Epidemiology and Public Health, Pasteur Institute of Lille, Lille, France. [58]Department of Public Health, Lille University Hospital, Lille, France. [59]Department of Psychiatry, University of California San Diego, San Diego, CA, USA. [60]Department of Psychological and Brain Sciences, Boston University, Boston, MA, USA. [61]NORMENT, KG Jebsen Centre for Psychosis Research, Institute of Clinical Medicine, University of Oslo and Division of Mental Health and Addiction, Oslo University Hospital, Oslo, Norway. [62]Departments of Radiology and Neurosciences, University of California, San Diego, La Jolla, CA, USA. [63]National Center for PTSD at Boston VA Healthcare System, Boston, MA, USA. [64]Department of Psychiatry and Department of Medicine-Biomedical Genetics Section, Boston University School of Medicine, Boston, MA, USA. [65]Psychiatric Genetics, QIMR Berghofer Medical Research Institute, Brisbane, QLD, Australia. [66]Imaging Genetics Center, Mark and Mary Stevens Neuroimaging and Informatics Institute, Keck School of Medicine of USC, University of Southern California, Los Angeles, CA, USA. [67]Department of Human Genetics, Radboud university medical center, Nijmegen, The Netherlands. [68]Donders Institute for Brain, Cognition and Behaviour, Radboud University, Nijmegen, The Netherlands. [69]Neuroscience Biomarkers, Janssen Research and Development, LLC, San Diego, CA, USA. [70]Department of Genetics & UNC Neuroscience Center, University of North Carolina at Chapel Hill, Chapel Hill, NC, USA. [71]Neuropsychiatric Institute, Prince of Wales Hospital, Sydney, NSW, Australia. [72]Day Clinic for Cognitive Neurology, University Hospital Leipzig, Leipzig, Germany. [73]Nuffield Department of Population Health, University of Oxford, Oxford, UK. [74]Departments of Neurology and Epidemiology, University of Washington, Seattle, WA, USA. [75]Bloorview Research Institute, Holland Bloorview Kids Rehabilitation Hospital, Toronto, ON, Canada. [76]Departments of Psychology and Psychiatry, University of Toronto, Toronto, ON, Canada. [77]CHU de Bordeaux, Department of Neurology, F-33000 Bordeaux, France. [78]Department of Neurology, Erasmus MC, Rotterdam, The Netherlands. [229]These authors contributed equally: Edith Hofer, Gennady V. Roshchupkin, Hieab H.H. Adams. [230]These authors jointly supervised this work: Reinhold Schmidt, Sudha Seshadri. *A list of authors and their affiliations appears at the end of the paper. ✉email: reinhold. schmidt@medunigraz.at; suseshad@bu.edu

## ENIGMA consortium

Katrina L. Grasby[79], Neda Jahanshad[80], Jodie N. Painter[79], Lucía Colodro-Conde[79], Janita Bralten[81,82], Derrek P. Hibar[80,83], Penelope A. Lind[79], Fabrizio Pizzagalli[80], Christopher R. K. Ching[80], Mary Agnes B. McMahon[80], Natalia Shatokhina[80], Leo C. P. Zsembik[84], Ingrid Agartz[85], Saud Alhusaini[86], Marcio A. A. Almeida[87], Dag Alnæs[85], Inge K. Amlien[88], Micael Andersson[89], Tyler Ard[80], Nicola J. Armstrong[88], Allison Ashley-Koch[90], Manon Bernard[91], Rachel M. Brouwer[92], Elizabeth E. L. Buimer[92], Robin Bülow[93], Christian Bürger[94], Dara M. Cannon[95], Mallar Chakravarty[96], Qiang Chen[97], Joshua W. Cheung[80], Baptiste Couvy-Duchesne[98], Anders M. Dale[99], Shareefa Dalvie[100], Tânia K. de Araujo[101],

Greig I. de Zubicaray[102], Sonja M. C. de Zwarte[92], Anouk den Braber[103], Nhat Trung Doan[85], Katharina Dohm[94], Stefan Ehrlich[104], Hannah-Ruth Engelbrecht[105], Susanne Erk[106], Chun Chieh Fan[107], Iryna O. Fedko[103], Sonya F. Foley[108], Judith M. Ford[109], Masaki Fukunaga[110], Melanie E. Garrett[90], Tian Ge[111], Sudheer Giddaluru[112], Aaron L. Goldman[97], Nynke A. Groenewold[100], Dominik Grotegerd[94], Tiril P. Gurholt[85], Boris A. Gutman[80], Narelle K. Hansell[113], Mathew A. Harris[114,115,116,117], Marc B. Harrison[80], Courtney C. Haswell[118], Michael Hauser[90], Stefan Herms[119], Dirk J. Heslenfeld[120], New Fei Ho[121], David Hoehn[122], Per Hoffmann[119], Laurena Holleran[95], Martine Hoogman[81], Jouke-Jan Hottenga[103], Masashi Ikeda[123], Deborah Janowitz[124], Iris E. Jansen[125], Tianye Jia[126], Christiane Jockwitz[127], Ryota Kanai[128], Sherif Karama[129], Dalia Kasperaviciute[130], Tobias Kaufmann[85], Sinead Kelly[131], Masataka Kikuchi[132], Marieke Klein[81], Michael Knapp[133], Annchen R. Knodt[134], Bernd Krämer[135], Max Lam[121], Thomas M. Lancaster[108], Phil H. Lee[111], Tristram A. Lett[106], Lindsay B. Lewis[129], Iscia Lopes-Cendes[101], Michelle Luciano[114,136], Fabio Macciardi[137], Andre F. Marquand[138], Samuel R. Mathias[139], Tracy R. Melzer[140], Yuri Milaneschi[141], Nazanin Mirza-Schreiber[122], Jose C. V. Moreira[142], Thomas W. Mühleisen[127], Bertram Müller-Myhsok[122], Pablo Najt[95], Soichiro Nakahara[137], Kwangsik Nho[143], Loes M. Olde Loohuis[144], Dimitri Papadopoulos Orfanos[145], John F. Pearson[146], Toni L. Pitcher[140], Benno Pütz[122], Anjanibhargavi Ragothaman[80], Faisal M. Rashid[80], Ronny Redlich[94], Céline S. Reinbold[119], Jonathan Repple[94], Geneviève Richard[85], Brandalyn C. Riedel[80], Shannon L. Risacher[143], Cristiane S. Rocha[101], Nina Roth Mota[81], Lauren Salminen[80], Arvin Saremi[80], Andrew J. Saykin[143], Fenja Schlag[147], Lianne Schmaal[148], Peter R. Schofield[149,150], Rodrigo Secolin[101], Chin Yang Shapland[147], Li Shen[151], Jean Shin[91], Elena Shumskaya[81], Ida E. Sønderby[85], Emma Sprooten[82], Lachlan T. Strike[113], Katherine E. Tansey[152], Alexander Teumer[153], Anbupalam Thalamuthu[154], Sophia I. Thomopoulos[80], Diana Tordesillas-Gutiérrez[155], Jessica A. Turner[156], Anne Uhlmann[100], Costanza Ludovica Vallerga[98], Dennis van der Meer[85], Marjolein M. J. van Donkelaar[81], Liza van Eijk[157], Theo G. M. van Erp[137], Neeltje E. M. van Haren[92], Daan van Rooij[138], Marie-José van Tol[158], Jan H. Veldink[159], Ellen Verhoef[147], Esther Walton[156], Mingyuan Wang[121], Yunpeng Wang[85], Joanna M. Wardlaw[114,115,116,117], Wei Wen[154], Lars T. Westlye[85], Christopher D. Whelan[80], Stephanie H. Witt[160], Katharina Wittfeld[161,124], Christiane Wolf[162], Thomas Wolfers[81], Clarissa L. Yasuda[163], Dario Zaremba[94], Zuo Zhang[164], Alyssa H. Zhu[80], Marcel P. Zwiers[138], Eric Artiges[165], Amelia A. Assareh[154], Rosa Ayesa-Arriola[166], Aysenil Belger[118], Christine L. Brandt[85], Gregory G. Brown[167], Sven Cichon[119], Joanne E. Curran[87], Gareth E. Davies[168], Franziska Degenhardt[169], Bruno Dietsche[170], Srdjan Djurovic[171], Colin P. Doherty[172], Ryan Espiritu[173], Daniel Garijo[173], Yolanda Gil[173], Penny A. Gowland[174], Robert C. Green[175], Alexander N. Häusler[176], Walter Heindel[177], Beng-Choon Ho[178], Wolfgang U. Hoffmann[153], Florian Holsboer[179], Georg Homuth[180], Norbert Hosten[93], Clifford R. Jack Jr[181], MiHyun Jang[173], Andreas Jansen[170], Knut Kolskår[163], Sanne Koops[92], Axel Krug[170], Kelvin O. Lim[182], Jurjen J. Luykx[183], Daniel H. Mathalon[184], Karen A. Mather[154], Venkata S. Mattay[97], Sarah Matthews[185], Jaqueline Mayoral Van Son[166], Sarah C. McEwen[167], Ingrid Melle[163], Derek W. Morris[173], Bryon A. Mueller[182], Matthias Nauck[186], Jan E. Nordvik[187], Markus M. Nöthen[169], Daniel S. O'Leary[178], Nils Opel[94], Marie-. Laure Paillère Martinot[165], G. Bruce Pike[188], Adrian Preda[137], Erin B. Quinlan[164], Varun Ratnakar[173], Simone Reppermund[154], Vidar M. Steen[112], Fábio R. Torres[101], Dick J. Veltman[141], James T. Voyvodic[118], Robert Whelan[189], Tonya White[190], Hidenaga Yamamori[191], Marina K. M. Alvim[163], David Ames[192,193], Tim J. Anderson[140], Ole A. Andreassen[85], Alejandro Arias-Vasquez[194], Mark E. Bastin[114,115,116,117], Bernhard T. Baune[195], John Blangero[87], Dorret I. Boomsma[103], Henry Brodaty[154], Han G. Brunner[81], Randy L. Buckner[196], Jan K. Buitelaar[138], Juan R. Bustillo[197], Wiepke Cahn[92], Vince Calhoun[198], Xavier Caseras[152], Svenja Caspers[199], Gianpiero L. Cavalleri[86], Fernando Cendes[163], Aiden Corvin[200], Benedicto Crespo-Facorro[166], John C. Dalrymple-Alford[201], Udo Dannlowski[94], Eco J. C. de Geus[103], Ian J. Deary[114,136], Norman Delanty[202],

Chantal Depondt[203], Sylvane Desrivières[164], Gary Donohoe[95], Thomas Espeseth[204], Guillén Fernández[138], Simon E. Fisher[147], Herta Flor[205], Andreas J. Forstner[169], Clyde Francks[147], Barbara Franke[81], David C. Glahn[139], Randy L. Gollub[206], Hans J. Grabe[161,124], Oliver Gruber[135], Asta K. Håberg[207], Ahmad R. Hariri[134], Catharina A. Hartman[208], Ryota Hashimoto[209], Andreas Heinz[106], Manon H. J. Hillegers[190], Pieter J. Hoekstra[208], Avram J. Holmes[139], L. Elliot Hong[210], William D. Hopkins[211], Hilleke E. Hulshoff Pol[92], Terry L. Jernigan[212], Erik G. Jönsson[213], René S. Kahn[214], Martin A. Kennedy[215], Tilo T. J. Kircher[170], Peter Kochunov[210], John B. J. Kwok[150,216], Stephanie Le Hellard[112], Nicholas G. Martin[79], Jean -. Luc Martinot[165], Colm McDonald[95], Katie L. McMahon[217], Andreas Meyer-Lindenberg[218], Rajendra A. Morey[118], Lars Nyberg[89], Jaap Oosterlaan[219], Roel A. Ophoff[144], Tomáš Paus[220,221], Zdenka Pausova[91,222], Brenda W. J. H. Penninx[141], Tinca J. C. Polderman[125], Danielle Posthuma[125], Marcella Rietschel[160], Joshua L. Roffman[206], Laura M. Rowland[210], Perminder S. Sachdev[154], Philipp G. Sämann[122], Gunter Schumann[164], Kang Sim[223], Sanjay M. Sisodiya[208], Jordan W. Smoller[111], Iris E. Sommer[224], Beate St Pourcain[185], Dan J. Stein[100], Arthur W. Toga[80], Julian N. Trollor[154,225], Nic J. A. Van der Wee[226], Dennis van 't Ent[103], Henry Völzke[153], Henrik Walter[106], Bernd Weber[227], Daniel R. Weinberger[97], Margaret J. Wright[113], Juan Zhou[228], Jason L. Stein[84], Paul M. Thompson[80] & Sarah E. Medland[79]

[79]Psychiatric Genetics, QIMR Berghofer Medical Research Institute, Brisbane, QLD, Australia. [80]Imaging Genetics Center, Mark and Mary Stevens Neuroimaging and Informatics Institute, Keck School of Medicine of USC, University of Southern California, Los Angeles, CA, USA. [81]Department of Human Genetics, Radboud university medical center, Nijmegen, The Netherlands. [82]Donders Institute for Brain, Cognition and Behaviour, Radboud University, Nijmegen, The Netherlands. [83]Neuroscience Biomarkers, Janssen Research and Development, LLC, San Diego, CA, USA. [84]Department of Genetics & UNC Neuroscience Center, University of North Carolina at Chapel Hill, Chapel Hill, NC, USA. [85]NORMENT, KG Jebsen Centre for Psychosis Research, Institute of Clinical Medicine, University of Oslo and Division of Mental Health and Addiction, Oslo University Hospital, Oslo, Norway. [86]Department of Molecular and Cellular Therapeutics, Royal College of Surgeons in Ireland, Dublin, Ireland. [87]Department of Human Genetics and South Texas Diabetes and Obesity Institute, Rio Grande Valley School of Medicine, , University of Texas, Brownsville, USA. [88]Centre for Lifespan Changes in Brain and Cognition, Department of Psychology, University of Oslo, Oslo, Norway. [89]Department of Integrative Medical Biology, Umeå University, Umeå, Sweden. [90]Duke Molecular Physiology Institute, Duke University Medical Center, Durham, NC, USA. [91]Hospital for Sick Children, Toronto, ON, Canada. [92]Department of Psychiatry, Brain Center Rudolf Magnus, University Medical Center Utrecht, Utrecht University, Utrecht, The Netherlands. [93]Institute for Diagnostic Radiology and Neuroradiology, University Medicine Greifswald, Greifswald, Germany. [94]Department of Psychiatry, , University of Münster, Münster, Germany. [95]Centre for Neuroimaging & Cognitive Genomics, National University of Ireland Galway, Galway, Ireland. [96]Douglas Mental Health University Institute, McGill University, Montreal, QC, Canada. [97]Lieber Institute for Brain Development, Baltimore, MD, USA. [98]Institute for Molecular Bioscience, The University of Queensland, Brisbane, QLD, Australia. [99]Departments of Radiology and Neurosciences, University of California, San Diego, La Jolla, CA, USA. [100]Department of Psychiatry and Mental Health, University of Cape Town, Cape Town, South Africa. [101]Department of Medical Genetics, School of Medical Sciences, University of Campinas - UNICAMP, Campinas, Brazil. [102]Faculty of Health, Institute of Health and Biomedical Innovation, Queensland University of Technology, Brisbane, QLD, Australia. [103]Department of Biological Psychology, Vrije Universiteit Amsterdam, Amsterdam, The Netherlands. [104]Division of Psychological & Social Medicine and Developmental Neurosciences, Technische Universität Dresden, Dresden, Germany. [105]Division of Human Genetics, Institute of Infectious Disease and Molecular Medicine,  University of Cape Town, Cape Town, South Africa. [106]Division of Mind and Brain Research, Department of Psychiatry and Psychotherapy, Campus Charité Mitte, Charité - Universitätsmedizin Berlin, Berlin, Germany. [107]Department of Cognitive Science, University of California San Diego, San Diego, CA, USA. [108]Cardiff University Brain Research Imaging Centre, Cardiff University, Cardiff, UK. [109]San Francisco Veterans Administration Medical Center, San Francisco, CA, USA. [110]Division of Cerebral Integration, National Institute for Physiological Sciences, Okazaki, Japan. [111]Psychiatric and Neurodevelopmental Genetics Unit, Center for Genomic Medicine, Massachusetts General Hospital, Boston, MA, USA. [112]NORMENT - K.G. Jebsen Centre for Psychosis Research, Department of Clinical Science, NORMENT University of Bergen, Bergen, Norway. [113]Queensland Brain Institute, The University of Queensland, St Lucia, QLD, Australia. [114]Centre for Cognitive Epidemiology and Cognitive Ageing, University of Edinburgh, Edinburgh, UK. [115]Centre for Clinical Brain Sciences, University of Edinburgh, Edinburgh, UK. [116]Brain Research Imaging Centre, University of Edinburgh, Edinburgh, UK. [117]Scottish Imaging Network, A Platform for Scientific Excellence (SINAPSE) Collaboration, Department of Neuroimaging Sciences, The University of Edinburgh, Edinburgh, UK. [118]Duke UNC Brain Imaging and Analysis Center, Duke University Medical Center, Durham, NC, USA. [119]Department of Biomedicine, University of Basel, Basel, Switzerland. [120]Department of Cognitive and Clinical Neuropsychology, Vrije Universiteit Amsterdam, Amsterdam, The Netherlands. [121]Research Division, Institute of Mental Health, Singapore, Singapore. [122]Max Planck Institute of Psychiatry, Munich, Germany. [123]Department of Psychiatry, Fujita Health University School of Medicine, Toyoake, Japan. [124]Department of Psychiatry and Psychotherapy, University Medicine Greifswald, Greifswald, Germany. [125]Complex Trait Genetics, Center for Neurogenomics and Cognitive Research, Vrije Universiteit Amsterdam, Amsterdam, The Netherlands. [126]Institute of Science and Technology for Brain-Inspired Intelligence, Fudan University, Shanghai, China. [127]Institute of Neuroscience and Medicine (INM-1), Research Centre Jülich, Jülich, Germany. [128]Department of Neuroinformatics, Araya, Inc, Tokyo, Japan. [129]McGill University, Montreal Neurological Institute, Montreal, QC, Canada. [130]Department of Clinical and Experimental Epilepsy, UCL Institute of Neurology, London, UK. [131]Public Psychiatry Division, Massachusetts Mental Health Center, Beth Israel Deaconess Medical Center, Harvard Medical School, Boston, MA, USA. [132]Department of Genome Informatics, Graduate School of Medicine, Osaka University, Suita, Japan. [133]Department of Medical Biometry, Informatics and Epidemiology, University Hospital Bonn, Bonn, Germany. [134]Department of Psychology and Neuroscience, Duke University, Durham, NC, USA. [135]Section for Experimental Psychopathology and Neuroimaging, Department of General Psychiatry,  Heidelberg University Hospital, Heidelberg, Germany. [136]Department of Psychology, University of Edinburgh, Edinburgh, UK. [137]Department of Psychiatry and Human Behavior, School of Medicine, University of California, Irvine, Irvine, CA, USA. [138]Department of Cognitive

Neuroscience, Radboud university medical center, Nijmegen, The Netherlands. [139]Department of Psychiatry, Yale University School of Medicine, New Haven, CT, USA. [140]Department of Medicine, , University of Otago, Christchurch, Christchurch, New Zealand. [141]Psychiatry, Amsterdam UMC Vrije Universiteit, Amsterdam, The Netherlands. [142]BRAINN - Brazilian Institute of Neuroscience and Neurotechnology, Campinas, Brazil. [143]Department of Radiology and Imaging Sciences, Indiana University School of Medicine, Indianapolis, IN, USA. [144]Center for Neurobehavioral Genetics, University of California Los Angeles, Los Angeles, CA, USA. [145]NeuroSpin, CEA, Université Paris-Saclay, Gif sur Yvette, France. [146]Biostatistics and Computational Biology Unit, University of Otago, Christchurch, Christchurch, New Zealand. [147]Language and Genetics Department, Max Planck Institute for Psycholinguistics, Nijmegen, The Netherlands. [148]Orygen, The National Centre of Excellence for Youth Mental Health, Melbourne, Australia. [149]Neuroscience Research Australia, Sydney, NSW, Australia. [150]School of Medical Sciences, University of New South Wales, Sydney, NSW, Australia. [151]Department of Biostatistics, Epidemiology and Informatics, University of Pennsylvania, Philadelphia, PA, USA. [152]MRC Centre for Neuropsychiatric Genetics and Genomics, Cardiff University, Cardiff, UK. [153]Institute for Community Medicine, University Medicine Greifswald, Greifswald, Germany. [154]Centre for Healthy Brain Ageing, School of Psychiatry, University of New South Wales, Sydney, NSW, Australia. [155]Neuroimaging Unit, Valdecilla Biomedical Research Institute IDIVAL, Santander, Spain. [156]Department of Psychology, Georgia State University, Atlanta, GA, USA. [157]School of Psychology, University of Queensland, Brisbane, QLD, Australia. [158]Cognitive Neuroscience Center, Department of Neuroscience, University Medical Center Groningen, Groningen, The Netherlands. [159]Department of Neurology, Brain Center Rudolf Magnus, University Medical Center Utrecht, Utrecht University, Utrecht, The Netherlands. [160]Department of Genetic Epidemiology in Psychiatry, Central Institute of Mental Health, Medical Faculty Mannheim, Heidelberg University, Mannheim, Germany. [161]German Center for Neurodegenerative Diseases (DZNE), Site Rostock/ Greifswald, Greifswald, Germany. [162]Department of Psychiatry, Psychosomatics and Psychotherapy, University of Würzburg, Würzburg, Germany. [163]Department of Neurology, FCM, University of Campinas - UNICAMP, Campinas, Brazil. [164]Social, Genetic and Developmental Psychiatry Centre, Institute of Psychiatry, , Psychology & Neuroscience, King's College London, London, UK. [165]INSERM Unit 1000 - Neuroimaging & Psychiatry, Paris Saclay University, Gif sur Yvette, France. [166]Department of Psychiatry, University Hospital Marqués de Valdecilla, School of Medicine, University of Cantabria–IDIVAL, Santander, Spain. [167]Department of Psychiatry, University of California San Diego, San Diego, CA, USA. [168]Avera Institute for Human Genetics, Sioux Falls, SD, USA. [169]Institute of Human Genetics, School of Medicine & University Hospital Bonn, University of Bonn, Bonn, Germany. [170]Department of Psychiatry and Psychotherapy, Philipps-University Marburg, Marburg, Germany. [171]Department of Medical Genetics, Oslo University Hospital, Oslo, Norway. [172]Department of Neurology, St James's Hospital, Dublin, Ireland. [173]Information Sciences Institute, University of Southern California, Los Angeles, CA, USA. [174]Sir Peter Mansfield Imaging Centre, University of Nottingham, Nottingham, UK. [175]Brigham and Women's Hospital, Boston, MA, USA. [176]Center for Economics and Neuroscience, University of Bonn, Bonn, Germany. [177]Department of Clinical Radiology, University of Münster, Münster, Germany. [178]Department of Psychiatry, University of Iowa College of Medicine, Iowa City, IA, USA. [179]HMNC Holding GmbH, Munich, Germany. [180]Interfaculty Institute for Genetics and Functional Genomics, University Medicine Greifswald, Greifswald, Germany. [181]Department of Radiology, Mayo Clinic, Rochester, MN, USA. [182]Department of Psychiatry, University of Minnesota, Minneapolis, MN, USA. [183]Department of Translational Neuroscience, Brain Center Rudolf Magnus, University Medical Center Utrecht, Utrecht University, Utrecht, The Netherlands. [184]Department of Psychiatry and Weill Institute for Neurosciences, University of California San Francisco, San Francisco, CA, USA. [185]MRC Integrative Epidemiology Unit, Department of Population Health Sciences, Bristol Medical School, Bristol, UK. [186]Institute of Clinical Chemistry and Laboratory Medicine, University Medicine Greifswald, Greifswald, Germany. [187]Sunnaas Rehabilitation Hospital HT, Nesodden, Norway. [188]Departments of Radiology and Clinial Neurosciences, University of Calgary, Calgary, AB, Canada. [189]School of Psychology, Trinity College Dublin, Dublin, Ireland. [190]Department of Child and Adolescent Psychiatry/Psychology, Erasmus Medical Center-Sophia Children's Hospital, Rotterdam, The Netherlands. [191]Department of Psychiatry, Osaka University Graduate School of Medicine, Suita, Japan. [192]National Ageing Research Institute, Royal Melbourne Hospital, Parkville, VIC, Australia. [193]Academic Unit for Psychiatry of Old Age, University of Melbourne, St George's Hospital, Kew, VIC, Australia. [194]Department of Psychiatry, Radboud university medical center, Nijmegen, The Netherlands. [195]Department of Psychiatry, The University of Melbourne, Melbourne, VIC, Australia. [196]Department of Psychology and Center for Brain Science, Harvard University, Boston, MA, USA. [197]Department of Psychiatry, University of New Mexico, Albuquerque, NM, USA. [198]Department of Electrical and Computer Engineering, The University of New Mexico, Albuquerque, NM, USA. [199]Institute for Anatomy I Medical Faculty, Heinrich-Heine University, Düsseldorf, Germany. [200]Department of Psychiatry, School of Medicine, Trinity College Dublin, Dublin, Ireland. [201]Department of Psychology, University of Canterbury, Christchurch, New Zealand. [202]FutureNeuro Research Centre, Royal College of Surgeons in Ireland, Dublin, Ireland. [203]Department of Neurology, Hôpital Erasme, Université Libre de Bruxelles, Brussels, Belgium. [204]Department of Psychology, University of Oslo, Oslo, Norway. [205]Department of Cognitive and Clinical Neuroscience, Central Institute of Mental Health, Medical Faculty Mannheim, Heidelberg University, Mannheim, Germany. [206]Department of Psychiatry, Massachusetts General Hospital, Boston, MA, USA. [207]Department of Neuroscience, Norwegian University of Science and Technology, Trondheim, Norway. [208]Department of Psychiatry, University Medical Center Groningen, University of Groningen, Groningen, The Netherlands. [209]Molecular Research Center for Children's Mental Development, United Graduate School of Child Development, Osaka University, Suita, Japan. [210]Department of Psychiatry, Maryland Psychiatry Research Center, University of Maryland School of Medicine, Baltimore, MD, USA. [211]Neuroscience Institute, Georgia State University, Atlanta, GA, USA. [212]Center for Human Development, University of California San Diego, La Jolla, CA, USA. [213]Centre for Psychiatric Research, Department of Clinical Neuroscience, Karolinska Institutet, Stockholm, Sweden. [214]Department of Psychiatry, Icahn School of Medicine at Mount Sinai, New York, NY, USA. [215]Department of Pathology and Biomedical Science, University of Otago, Christchurch, Christchurch, New Zealand. [216]Brain and Mind Centre - The University of Sydney, Camperdown, NSW, Australia. [217]Herston Imaging Research Facility, School of Clinical Sciences, Queensland University of Technology, Brisbane, QLD, Australia. [218]Department of Psychiatry and Psychotherapy, Central Institute of Mental Health, Medical Faculty Mannheim, Heidelberg University, Mannheim, Germany. [219]Emma Children's Hospital, Academic Medical Center, Amsterdam, The Netherlands. [220]Bloorview Research Institute, Holland Bloorview Kids Rehabilitation Hospital, Toronto, ON, Canada. [221]Departments of Psychology and Psychiatry, University of Toronto, Toronto, ON, Canada. [222]Departments of Physiology and Nutritional Sciences, University of Toronto, Toronto, ON, Canada. [223]General Psychiatry, Institute of Mental Health, Singapore, Singapore. [224]Department of Medical and Biological Psychology, University of Bergen, Bergen, Norway. [225]Department of Developmental Disability Neuropsychiatry, School of Psychiatry, University of New South Wales, Sydney, NSW, Australia. [226]Department of Psychiatry, Leiden University Medical Center, Leiden, The Netherlands. [227]Institute of Experimental Epileptology and Cognition Research, University Hospital Bonn, Bonn, Germany. [228]Center for Cognitive Neuroscience, Neuroscience and behavioral disorders program, Duke-National University of Singapore Medical School, Singapore, Singapore.

