## [Peer Review File · Nature Communications]

REVIEWERS' COMMENTS:

Reviewer #2 (Remarks to the Author):

The manuscript under review is "Genetic Determinants of Cortical Structure (Thickness, Surface Area and Volumes) in 2 General Population Samples of 22,824 Adults". The authors describe a meta analysis of many consortia, providing details for the mapping between genotype and brain phenotype.

This work is of great quality, and many improvements have been made since it was last reviewed by Nature Group, as follows.

1) More conservative thresholds for multiple test correction are now considered (in response to remark 1 of Reviewer #1). The authors note that BF correction reduces significant findings from 160 to 142. They now mention that BF destroys the CTh signal in the main text. 142 instead of 160 associations is still a valuable contribution to the effort to detail the mapping from genotype to brain phenotype. The authors may consider adding the 142 number to the main text or add a BF column to a table, to share this more standard analysis.

2) The authors provide an excellent analysis of co-heritability between the phenotypes they consider and brain-related traits. In particular, p-values in Table S16 for traits such as Parkinson's show more significance than in other studies such as Elliott et al. While still not significant at a BF level, this is a direction required in order to make progress in multiphenotype analysis. Also, the summarization of the genetic covariance in Figure S15 is much more clearly displayed and useful to researchers than the reports in Elliott et al. and other related work.

Minor point:

1) Page 25 line 11 "As recommended by the ldsc tool developers" -> "As recommended by the LDSC tool developers"

Reviewer #3 (Remarks to the Author):

This reviewer is satisfied with most responses from the authors. However, I still have a minor concern about the possible population structure and its potential consequences in this study. Specifically, this study integrates datasets from multiple different studies with different age range and MRI acquisition protocols. It seems that the authors ignored the heterogeneous population structure. This needs some discussions.

In addition, since the neuroimaging measures show a polygenic genetic architecture (i.e., many contributing loci are founded across the genome), it is of great interest to perform polygenic risk scores prediction (e.g., Vilhjálmsson et al., 2015 <https://doi.org/10.1016/j.ajhg.2015.09.001>) to check the out-of-sample genetics prediction power of these brain imaging phenotypes.

REVIEWERS' COMMENTS:

Reviewer #2 (Remarks to the Author):

The manuscript under review is "Genetic Determinants of Cortical Structure (Thickness, Surface Area and Volumes) in 2 General Population Samples of 22,824 Adults". The authors describe a meta analysis of many consortia, providing details for the mapping between genotype and brain phenotype.

This work is of great quality, and many improvements have been made since it was last reviewed by Nature Group, as follows.

1) More conservative thresholds for multiple test correction are now considered (in response to remark 1 of Reviewer #1). The authors note that BF correction reduces significant findings from 160 to 142. They now mention that BF destroys the CTh signal in the main text. 142 instead of 160 associations is still a valuable contribution to the effort to detail the mapping from genotype to brain phenotype. The authors may consider adding the 142 number to the main text or add a BF column to a table, to share this more standard analysis.

Response:

We agree with the reviewer and now write in **Results / Genome-wide association analysis (page 9, lines 20-22)**:

"If we had used a more stringent threshold of $p_{\text{discovery}} < 4.76 \times 10^{-10} = 5 \times 10^{-8} / 105$, correcting for all the 105 GWAS analyses performed, we would have identified 142 significant associations (Supplementary Tables 1-4)."

In Supplementary Tables 1-4, discovery p-values $< 4.76 \times 10^{-10}$ ($= 5 \times 10^{-8} / 105$) are now shown in bold.

2) The authors provide an excellent analysis of co-heritability between the phenotypes they consider and brain-related traits. In particular, p-values in Table S16 for traits such as Parkinson's show more significance than in other studies such as Elliott et al. While still not significant at a BF level, this is a direction required in order to make progress in multiphenotype analysis. Also, the summarization of the genetic covariance in Figure S15 is much more clearly displayed and useful to researchers than the reports in Elliott et al. and other related work.

Response:

We thank the reviewer for this encouraging comment!

Minor point:

1) Page 25 line 11 "As recommended by the Idsc tool developers" -> "As recommended by the LDSC tool developers"

Response:

We have changed "Idsc" to LDSC in the manuscript.

Reviewer #3 (Remarks to the Author):

This reviewer is satisfied with most responses from the authors. However, I still have a minor concern about the possible population structure and its potential consequences in this study. Specifically, this study integrates datasets from multiple different studies with different age range and MRI acquisition protocols. It seems that the authors ignored the heterogeneous population structure. This needs some discussions.

Response:

We now have now changed the “limitations” paragraph in the **Discussion (page 19, lines 23-25 and page 20, lines 1-5):**

“A limitation of our study is the heterogeneity of the MR phenotypes between cohorts due to different scanners, field strengths, MR protocols and MRI analysis software. This heterogeneity as well as the different age ranges in the participating cohorts may have caused different effects over the cohorts. We nevertheless combined the data of the individual cohorts to maximize the sample size as it has been done in previous CHARGE GWAS analyses³¹⁻³³. To account for the heterogeneity we used a sample-size weighted meta-analysis which does not provide overall effect estimates. This method has lower power to detect associations compared to inverse-variance weighted meta-analysis and we therefore may have found less associations.”

31 van der Lee, S. J. *et al.* A genome-wide association study identifies genetic loci associated with specific lobar brain volumes. *Commun Biol* 2, 285, doi:10.1038/s42003-019-0537-9 (2019).

32 Ikram, M. A. *et al.* Common variants at 6q22 and 17q21 are associated with intracranial volume. *Nat Genet* 44, 539-544, doi:10.1038/ng.2245 (2012).

33 Adams, H. H. *et al.* Novel genetic loci underlying human intracranial volume identified through genome-wide association. *Nat Neurosci* 19, 1569-1582, doi:10.1038/nn.4398 (2016).

*In addition, since the neuroimaging measures show a polygenic genetic architecture (i.e., many contributing loci are founded across the genome), it is of great interest to perform polygenic risk scores prediction (e.g., Vilhjálmsson *et al.*, 2015 <https://doi.org/10.1016/j.ajhg.2015.09.001>) to check the out-of-sample genetics prediction power of these brain imaging phenotypes.*

Response:

We thank the reviewer for this suggestion. Indeed, PRS analysis is an important utilization of the GWAS results. We performed PRS analysis by applying PRSice-2 (<https://www.prsice.info/>) to 7800 out-of-sample subjects (not included in the current GWAS analysis) from the UK Biobank. PRSice-2⁵⁷ is a software for automated PRS analyses on large scale data, and is therefore very well suited for the large number of phenotypes in our study. We provide the results of the PRS analysis in Supplementary Table 9 and now write in **Results / Associations across cortical measures and with other traits (page 10, lines 24-25 and page 11, lines 1-4):**

“Out-of-sample polygenic risk score (PRS) analyses showed associations ($p_{PRS} < 4.76 \times 10^{-3}$) with all investigated cortical measures in all cortical regions in 7800 UK Biobank individuals (Supplementary Table 9). For CTh, we

observed the maximum phenotypic variance explained by the PRS (R_{PRS}^2) in the global cortex ($R_{\text{PRS}}^2=0.015$, $p_{\text{PRS}}=1.05 \times 10^{-26}$), and for CSA and CV in the pericalcarine cortex ($R_{\text{PRS}}^2, \text{CSA}=0.029$, $p_{\text{PRS,CSA}}=1.29 \times 10^{-50}$; $R_{\text{PRS}}^2, \text{CV}=0.032$, $p_{\text{PRS,CV}}=5.30 \times 10^{-56}$).”

We also added the following description to **Methods / Genome-wide association analysis (page 23, line 25 and page 24, lines 1-4)**:

“PRS analysis was performed for 7800 out of sample subjects (not included in the current GWAS) from UK Biobank cohort using the PRSice-2 software⁵⁷ with standard settings. The significance threshold for the association between the PRS and the phenotype was set to 4.76×10^{-3} ($=0.05/105$: nominal significance divided by number GWAS phenotypes). “

⁵⁷ Choi, S. W. & O'Reilly, P. F. PRSice-2: Polygenic Risk Score software for biobank-scale data. *Gigascience* 8, doi:10.1093/gigascience/giz082 (2019).